# CARTRIDGES: LIGHTWEIGHT AND GENERAL-PURPOSE LONG CONTEXT REPRESENTATIONS VIA SELF-STUDY

**Sabri Eyuboglu**[1*]    **Ryan Ehrlich**[1*]    **Simran Arora**[1,2*]    **Neel Guha**[1]    **Dylan Zinsley**[3]    **Emily Liu**[1]
**Atri Rudra**[3]    **James Zou**[1]    **Azalia Mirhoseini**[1]    **Christopher Ré**[1]

[1]Stanford University    [2]Caltech    [3]University at Buffalo    *Equal contribution
eyuboglu@stanford.edu

## ABSTRACT

Large language models are often used to answer queries grounded in large text corpora (*e.g.* codebases, legal documents, or chat histories) by placing the entire corpus in the context window and leveraging in-context learning (ICL). Although current models support contexts of 100K–10M tokens, this setup is costly to serve because the memory consumption of the KV cache scales with input length. We explore an alternative: training a smaller KV cache offline on each corpus. At inference time, we load this trained KV-cache, which we call a CARTRIDGE, and decode a response. Critically, the cost of training a CARTRIDGE can be amortized across all the queries referencing the same corpus. However, we find that the naive approach of training the CARTRIDGE with next-token prediction on the corpus is not competitive with ICL. Instead, we propose SELF-STUDY, a training recipe in which we generate synthetic conversations about the corpus and train the CARTRIDGE with a context-distillation objective. We find that CARTRIDGES trained with SELF-STUDY replicate the functionality of ICL, while being significantly cheaper to serve. On challenging long-context benchmarks, CARTRIDGES trained with SELF-STUDY match ICL performance while using $38.6\times$ less memory on average and enabling $26.4\times$ higher throughput. SELF-STUDY also extends the model's effective context length (*e.g.* from 128k to 484k tokens on MTOB) and surprisingly, leads to CARTRIDGES that can be composed at inference time without retraining.

## 1 INTRODUCTION

Large language model (LLM) users often place large text corpora into the context window. For instance, a user or organization may use LLMs to understand a codebase (Nam et al., 2024), financial document (Islam et al., 2023), legal texts (Guha et al., 2023), a textbook (Ouellette et al., 2025), or personal files (Arora & Ré, 2022). LLMs excel here due to in-context learning (ICL), enabling accurate responses to diverse queries (e.g., questions, summarization, reasoning) (Dong et al., 2022).

Despite its flexibility, this usage paradigm is costly to serve. ICL requires maintaining a KV cache that grows linearly with the input length. For example, LLaMA 70B needs 84 GB of memory (at 16-bit precision) to answer a single question over a 128k-token context (Dubey et al., 2024). This severely limits user throughput: on a single H100 GPU, LLaMA 8B's peak throughput (tokens/s) drops by $77\times$ when increasing the context from 1k to 120k tokens (Figure 2).

Prior work has thus explored ways to reduce KV cache memory usage. For instance, prompt compression methods reduce the number of tokens stored in the cache via summarization, or self-information filtering (Jiang et al., 2023b; Li, 2023; Chuang et al., 2024), while KV cache compression techniques directly compress the stored key-value pairs (Ge et al., 2023a; Zhang et al., 2023b; Tang et al., 2024; Oren et al., 2024). Unfortunately, there are memory-quality tradeoffs associated with these methods: in experiments on challenging long-context tasks, we find that performance degrades rapidly when applying these methods with compression ratios greater than $2\times$ (see Figure 3).

Motivated by the observation that the cost of preparing a KV cache can be amortized across many queries that reference the same corpus, we explore a complementary approach based on offline training. Given a specific text corpus (*e.g.* a patient's medical record) we freeze the LLM and train

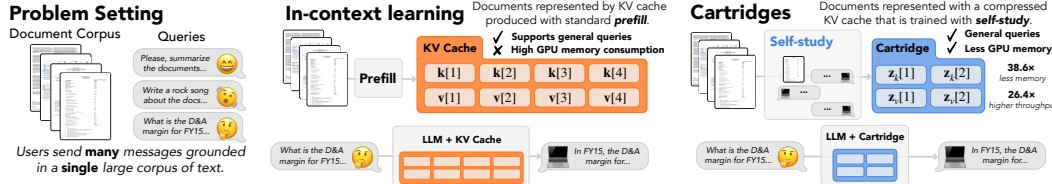

Figure 1: **Producing CARTRIDGES via self-study**. For a given document corpus, we train a CARTRIDGE by distilling the corpus into a parameterized KV cache through a process we call SELF-STUDY. At inference time, this CARTRIDGE can be loaded into an LLM, which can then be used to answer diverse queries about the corpus while requiring substantially less memory.

a smaller KV cache offline by backpropagating loss into the key and value vectors in a process equivalent to prefix tuning (Li & Liang, 2021; Lester et al., 2021). We call the trained KV cache representing the corpus a "CARTRIDGE." At inference time, we load the trained CARTRIDGE, append the user's messages, and decode. Because users repeatedly reference the same corpora (*e.g.* SEC filings, codebase, personal files), each CARTRIDGE can be trained once offline and reused. This approach also integrates cleanly with existing inference servers, which are already designed to manage per-user KV caches (Kwon et al., 2023; Zheng et al., 2024; Juravsky et al., 2025; Ye et al., 2025).

The central challenge of this work lies in training CARTRIDGES that exhibit the **generality** of ICL. Due to ICL, a standard KV cache is a remarkably general-purpose, albeit large, representation of a corpus: a single cache can support diverse interactions from answering factual questions to writing poems (Dong et al., 2022). In contrast, naïvely training a CARTRIDGE with next-token prediction on the raw corpus yields compact but restricted representations of the corpus. With next-token prediction, we show we can memorize the corpus perfectly using a CARTRIDGE with $107\times$ less memory than the standard KV-cache. However, the CARTRIDGE is not a general-purpose representation – it can only regurgitate the corpus, not answer diverse queries (Figure 2). The challenge is to reduce memory consumption while maintaining generality.

To address this challenge and produce general-purpose *and* compact CARTRIDGES, we propose an automated method called SELF-STUDY. SELF-STUDY has two steps:

1. **Synthetic data generation** (Section 4.1): We generate synthetic training data by prompting the model to quiz itself about the corpus content. Training on the resulting conversations lets us avoid training on the same exact text multiple times and improves generality (see Figure 2). To support corpora that exceed the effective context length of the model, we chunk the corpus when generating synthetic conversations. We also curate a set of seed prompts that bias the synthetic conversations towards global reasoning, improving structural awareness (see Figure 4).

2. **Context distillation** (Section 4.2): We train on the synthetic conversations using a context-distillation objective (Bhargava et al., 2024; Snell et al., 2022), which aligns the CARTRIDGE-augmented model's next-token distributions with the distributions of the model with the corpus in context. We find that the context distillation substantially improves the quality of the CARTRIDGES compared to next-token-prediction (see Figure 4 center).

In summary, given a large corpus of text, our goal is to train a small virtual KV cache, termed CARTRIDGE, that when used by the model, mimics the conversational behavior of the model with the entire corpus in context. To do this, we generate synthetic conversations and train the CARTRIDGE on them with a context distillation objective — a recipe we call SELF-STUDY.

**Evaluations.** We evaluate CARTRIDGES trained with SELF-STUDY on a set of challenging benchmarks that pair a single large text corpus (100k-484k tokens) with a diverse set of queries (Islam et al., 2023; Adams et al., 2024; Tanzer et al., 2023). We make three claims. **First**, SELF-STUDY expands the quality-memory frontier—averaged across the benchmarks, CARTRIDGES produced with SELF-STUDY match ICL generality and quality while consuming $38.6\times$ less memory, enabling a $26.4\times$ increase in peak throughput (tokens/s) when serving many users with different corpora. These memory reductions and speedups represent an order of magnitude improvement over state-of-the-art cache compression baselines (*e.g.* DuoAttention (Xiao et al., 2024b)). **Second**, CARTRIDGES enables context length extrapolation. On the MTOB benchmark (Tanzer et al., 2023), where models must translate from Kalamang, a low-resource language, into English, we use SELF-STUDY with LLAMA-

8B to construct a small CARTRIDGE from a 484k token textbook. This CARTRIDGE outperforms ICL over the first $130,000$ tokens of the textbook by 11.0 chrF points and matches the ICL performance over a curated subset of the textbook. **Third**, SELF-STUDY also yields CARTRIDGES that can be composed without joint optimization: when we concatenate two CARTRIDGES the model can answer queries requiring knowledge from both (see Figure 5).

Additionally, we ablate the design decisions in SELF-STUDY and CARTRIDGES (Section 5.3 and Section A). Notably, we compare CARTRIDGES parameterized as a KV cache (Li & Liang, 2021) with CARTRIDGES parameterized as a LoRA (Hu et al., 2022) and find that KV cache parameterization performs better on both in-domain and out-of-domain tasks.

In this work, we demonstrate how we can reduce memory consumption during language model serving by scaling offline training compute. We hope this new axis of scaling will enable new applications that are currently bottlenecked by KV cache memory consumption, like coding agents with full-repository context or long-term memory in chatbots.

## 2 PRELIMINARIES

We begin by discussing related work (Section 2.1), formalizing our problem (Section 2.2), and providing background on language models and KV caches (Section 2.3).

### 2.1 RELATED WORK

*See Appendix B for a more comprehensive discussion of prior work.*

**Parameter Efficient Fine-Tuning and Knowledge Injection** In order to adapt a language model to a specific task or domain, practitioners commonly train a small number of parameters (usually a low-rank adapter), which augment or modify the original model (Hu et al., 2022; Li & Liang, 2021; Lester et al., 2021; Meng et al., 2024; Zaken et al., 2021). In our work, we build upon a less popular technique, prefix-tuning (Li & Liang, 2021; Lester et al., 2021), where we optimize internal activations for a set of "virtual" tokens preceding the input. Recent works on *knowledge injection* apply LoRA (or variants (Mao et al., 2025)) to store a text corpus in a small number of parameters (Zhang et al., 2023a; Xiao et al., 2023; Kujanpää et al., 2024; Mao et al., 2025; Kuratov et al., 2025; Su et al., 2025; Caccia et al., 2025). In contrast to our work, these papers do not focus on memory reductions or throughput improvements enabled by knowledge injection and do identify the importance of the prefix-tuning parameterization.

**Prompt and KV-cache compression** Many works have proposed techniques to reduce the size of the KV cache. One set of approaches focuses on making the prompt smaller—explicit methods alter the prompt text through summarization and filtering (Jiang et al., 2023b; Li, 2023; Chuang et al., 2024; Zhang et al., 2024b; Pan et al., 2024), while implicit methods compress prompt representations into a set of "soft" tokens (Chevalier et al., 2023; Yen, 2024; Ge et al., 2023b; Mu et al., 2023; Qin et al., 2023; Lester et al., 2021). Another set of approaches exploits observations about the structure of the KV cache (Yu et al., 2024; Chang et al., 2024; Kim et al., 2024) to drop (Ge et al., 2023a; Zhang et al., 2023b; Tang et al., 2024; Oren et al., 2024; Li et al., 2024b) or merge tokens (Wang et al., 2024; Zhang et al., 2024d; Wan et al., 2024).

**Architectural changes** A large body of work has studied architectural changes to the original multi-head attention operation (Vaswani et al., 2017) with the aim of reducing the memory footprint of the KV cache or replacing it with a memory object of constant size (*inter alia* Zaheer et al. (2020); Shazeer (2019); Liu et al. (2024a); Gu & Dao (2023); Behrouz et al. (2024). In Section E, we provide a theoretical analysis comparing CARTRIDGES with linear attention, one such architecture with constant memory footprint. Unlike SELF-STUDY and the compression approaches discussed above, which can be readily applied to any pre-trained Transformer, these architectural changes typically require retraining the model from scratch or using complex architecture conversion techniques (Zhang et al., 2024a).

### 2.2 PROBLEM SETUP

We assume a setting in which users issue a stream of diverse queries about a common corpus of text. We denote the corpus as $\mathcal{C}$ and the query set as $Q = \{q_1, q_2, \ldots, q_m\}$. For example, $\mathcal{C}$ may

correspond to the 2022 Form 10-K filing for AMD, which is almost 100k tokens. Analyst might as diverse queries with respect to this filing, including: (1) recalling factual information, (2) performing mathematical reasoning, or (3) even generating creative responses (e.g., a poem). Other illustrative examples of $\mathcal{C}$ include legal filings, code repositories, chat histories, and medical records.

Let $R = \{r_1, r_2, \ldots, r_m\}$ denote the responses the LLM produces for the queries. We have two objectives. First, we wish to maximize the quality of responses $R$ under some quality metric (*e.g.* accuracy). Second, we wish to minimize the LLM's memory footprint while it is answering questions with respect to the document. This is because larger memory footprints decrease throughput and necessitate more hardware to serve the same number of users (Figure 2, Right).

## 2.3 LANGUAGE MODELS AND KV CACHES

Recall that an LLM $\mathcal{F}$ accepts as input a sequence of $N$ tokens $\mathbf{x} \in \mathcal{V}^n$ drawn from a discrete vocabulary $\mathcal{V} \subset \mathbb{Z}$ of tokens, each represented by a unique integer. The output, which we denote $\mathcal{F}(\cdot|\mathbf{x})$, corresponds to a categorical distribution over a vocab $\mathcal{V}$ conditioned on the prefix $\mathbf{x} \in \mathcal{V}^n$. Inside the language model, each token $x[i]$ in $\mathbf{x}$ is embedded into a $d$-dimensional space, yielding a matrix $\mathbf{u} \in \mathbb{R}^{n \times d}$. The matrix $\mathbf{u}$ is passed through a stack of $L$ model layers, which each mix the matrix along the $n$ and $d$ dimensions, with layer $\ell$ outputting $\mathbf{y}^l \in \mathbb{R}^{n \times d}$. The final $\mathbf{y}^L$ is mapped to the logits over $\mathcal{V}$ with a linear projection.

Most modern language models use the self-attention operator (Vaswani et al., 2017). Given an input $\mathbf{u} \in \mathbb{R}^{n \times d}$ for sequence length $n$ and embedding dimension $d$, it computes the output $\mathbf{y}^l \in \mathbb{R}^{n \times d}$ via the softmax $\mathbf{y}[i] = \sum_{j=1}^{i} \frac{\exp(\mathbf{q}[i]^\top \mathbf{k}[j]/\sqrt{d})\mathbf{v}[j]}{\sum_{t=1}^{i} \exp(\mathbf{q}[i]^\top \mathbf{k}[t]/\sqrt{d})}$ over projections $\mathbf{q}, \mathbf{k}, \mathbf{v} = \mathbf{u}\mathbf{W}_q, \mathbf{u}\mathbf{W}_k, \mathbf{u}\mathbf{W}_v$. where weight matrices $\boldsymbol{W}_q$, $\boldsymbol{W}_k$ and $\boldsymbol{W}_v$ for each layer are learned during training.

We generate text from $\mathcal{F}$ one token at a time by sampling from $\mathcal{F}(\cdot \mid \mathbf{x})$ and appending the sampled token to $\mathbf{x}$. Critically, the attention operator is causal: every output $\mathbf{y}[i]$ is conditioned on prior tokens. This means we can store the keys and values for the prior tokens in a KV cache $\{\mathbf{k}[j], \mathbf{v}[j]\}_{j=1}^{i}$, which grows in $i$. Thus, generation proceeds in two phases: (1) *prefill*, where we compute the KV cache for the initial prompt $\mathbf{x}$ and (2) *decode*, where we generate the response token by token and append to the cache. The KV cache effectively serves as a representation of the corpus $\mathcal{C}$.

# 3 THE CARTRIDGE PARADIGM

In this section, we describe the CARTRIDGE paradigm, in which we generate representations of the corpus $\mathcal{C}$ offline with training, instead of constructing them on-the-fly with prefill.

## 3.1 FORMALIZING CARTRIDGES

Our goal is to train a CARTRIDGE for a given corpus $\mathcal{C}$. A CARTRIDGE is a small set of parameters $Z \in \mathbb{R}^*$ (*i.e.* an adapter (Li & Liang, 2021; Hu et al., 2022)) that augments an LLM $\mathcal{F}$ and causes it to behave as if it had $\mathcal{C}$ in its context window. Formally, let $\mathcal{F}_Z(\cdot|q)$ denote the distribution of $\mathcal{F}$ augmented with $Z$ given a query $q$. For all $q \in Q$, we want to ensure that samples $r_Z \sim \mathcal{F}_Z(\cdot|q)$ are as good or better than the ICL sample $r_q \sim \mathcal{F}(\cdot|\mathcal{C} \oplus q)$, according to some query-specific scoring function. Because $Q$ might span a diverse range of question types (e.g., mathematical reasoning, factual recall comprehension, summarization, and more), it is essential that $\mathcal{F}_Z$ can **generalize** across different $q \in Q$. This is non-trivial because $Q$ is unknown when $Z$ is being learned offline.

## 3.2 PARAMETERIZING CARTRIDGES

We parameterize $Z$ using prefix-tuning (Li & Liang, 2021). Specifically, we allocate a KV cache composed of *trainable* key and value vectors $\mathbf{z}_k, \mathbf{z}_v \in \mathbb{R}^{p \times d}$. The size of the full $Z \in \mathbb{R}^{L \times p \times d \times 2}$ is controlled by the hyperparameter $p$.

In ICL, the KV cache for $\mathcal{F}_\mathcal{C}(q)$ (where $\mathcal{C}$ is of length $n_\mathcal{C}$ and $Q$ is of length $n_Q$) would contain $n_\mathcal{C} + n_Q$ key-value pairs, with the first $n_\mathcal{C}$ corresponding to $\mathcal{C}$ and the last $n_Q$ corresponding to $Q$:

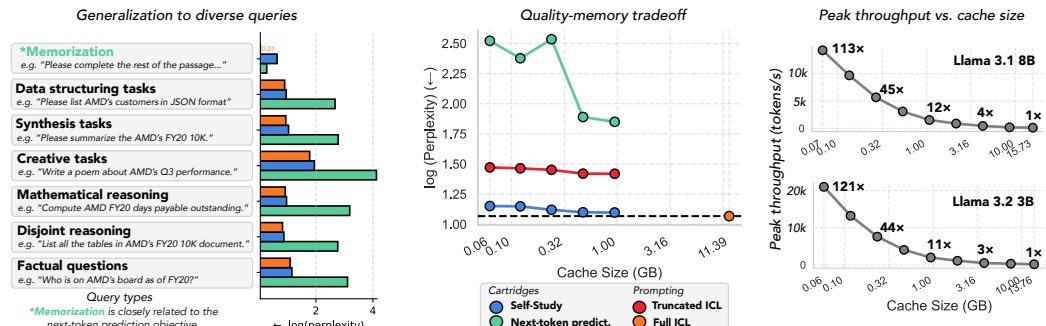

Figure 2: **CARTRIDGES trained with SELF-STUDY balance the generality and memory consumption tradeoff.** (**Left**) We evaluate on different slices from the GENCONVO dataset. CARTRIDGES trained with next-token prediction performs well on memorization queries, which resemble its training distribution, but cannot generalize to other queries like the other methods. (**Center**) The $x$-axis measures the size of the KV cache in GB for the different methods. The $y$-axis shows log-perplexity on the GENCONVO dataset averaged over the query types. (**Right**) Peak throughput (tokens/s) measured for different cache sizes for LLAMA-3B and LLAMA-8B with SGLang (Zheng et al., 2024) on an 1xH100 (See Section A).

$$
\underbrace{(\mathbf{k}_1, \mathbf{v}_1), \ldots, (\mathbf{k}_{n_\mathcal{C}}, \mathbf{v}_{n_\mathcal{C}})}_{\text{KV pairs for } \mathcal{C}}, \underbrace{(\mathbf{k}_{n_\mathcal{C}+1}, \mathbf{v}_{n_\mathcal{C}+1}) \ldots}_{\text{KV pairs for } q}
$$

**ICL KV Cache**

$$
\underbrace{(\mathbf{z}_1^k, \mathbf{z}_1^v), \ldots, (\mathbf{z}_p^k, \mathbf{z}_p^v)}_{\text{Trainable KV pairs in } Z}, \underbrace{(\mathbf{k}_{n_\mathcal{C}+1}, \mathbf{v}_{n_\mathcal{C}+1}) \ldots}_{\text{KV pairs for } q}
$$

**CARTRIDGE KV Cache**

To train a CARTRIDGE, we substitute the key-value pairs corresponding to $\mathcal{C}$ with $Z$, and directly optimize them by back-propagating the loss into the key and value vectors. **We freeze all model parameters, only training the keys and values in $Z$.** We discuss the choice of loss in Section 4.2.

**Initialization**    Prior work finds that optimizing a randomly initialized cache $Z$ is unstable and leads to degraded performance (Li & Liang, 2021). Instead, these works initialize the trainable cache with a smaller dimensionality $d$ and then re-project it to the original dimension with an MLP. In contrast, we find that proper initialization of $Z$ allows us to directly optimize the full cache without reparametrization. Specifically, we initialize $Z$ to the KV cache corresponding to the first $p$ tokens of the corpus $\mathcal{C}$. Alternatively, we could use a summary of the corpus or filter tokens using off-the-shelf prompt compression strategies (Xiao et al., 2024b). In Section 5.3, we show that our initializations lead to stable training and faster convergence than the random initialization.

*Why this parameterization?* We note that the parameter-efficient fine-tuning literature provides other ways to augment an LLM with a set of additional parameters, in particular low-rank adaptation (LoRA) (Li & Liang, 2021; Hu et al., 2022; Lester et al., 2021). In Section 5.3, we perform a comprehensive comparison of CARTRIDGES parameterized with prefix-tuning and LoRA.

### 3.3    SERVING CARTRIDGES

A CARTRIDGE can be served efficiently with minimal changes to existing LLM inference servers (Zheng et al., 2024; Kwon et al., 2023; Juravsky et al., 2025). Because a CARTRIDGE is a KV cache, it can be loaded directly into the KV cache managers of existing inference servers. LLM inference servers are heavily optimized for managing distinct KV-caches for multiple users (Ye et al., 2025), meaning CARTRIDGES can be served at high throughput using existing inference servers. Decoding tokens with a CARTRIDGE is identical to serving a request with a prefix of length $p$ (the hyperparameter denoting the number of tokens in the CARTRIDGE). This contrasts with other methods like LoRA, which require custom infrastructure to serve efficiently to multiple users (Chen et al., 2024a). See Figure 2 for the relationship between prefix length and throughput.

# 4 SELF-STUDY: A SELF-SUPERVISED METHOD FOR TRAINING CARTRIDGES

In this section, we describe SELF-STUDY, a simple approach for training a CARTRIDGE $Z$ on any corpus of text. The design of SELF-STUDY is motivated by the observation that CARTRIDGES trained with a simpler recipe fail to generalize to diverse user queries.

**Motivating observation**   The naive method for constructing a CARTRIDGE would be to fine-tune the parameters of $Z$ with the next token prediction objective on the corpus text directly. We show results experimenting with this approach in Figure 2, where we evaluate on a dataset derived from FinanceBench (Islam et al., 2023), which we refer to as GENCONVO (see Section D for details). GENCONVO contains multiple types of questions (*e.g.* synthesis, reasoning). We find that the naïve next-token prediction approach can memorize with near perfect perplexity (Figure 2 left), while consuming $107\times$ less memory than ICL (Figure 2 center). However, generalization to other slices is poor, as shown in Figure 2. We seek a training objective that allows the responses from a model that uses the CARTRIDGE to generalize to a diverse set of user queries, resembling ICL.

Motivated by these observations, we describe a synthetic data generation recipe in Section 4.1 and a context-distillation objective in Section 4.2. As we show in Figure 2, CARTRIDGES trained with this approach can generate responses to many types of queries that match the quality of queries generated with ICL. See Figure 1 for a visualization of the CARTRIDGE approach.

## 4.1 SELF-SUPERVISED SYNTHETIC DATA TO AVOID OVERFITTING

To improve CARTRIDGE generality, we propose generating a synthetic training dataset $\mathcal{D}_{\text{train}}$.

**Overall synthetic data pipeline**   Our overall pipeline puts information from the corpus $\mathcal{C}$ in context and prompts the model to have a conversation with itself about the corpus to generate the synthetic query-response pairs as shown in Algorithm 1. We represent the concatenation with $x \oplus y$.

---

**Algorithm 1** SELF-STUDY: Data Generation

**Input:** $\mathcal{C}$ : Corpus, $\mathcal{F}$ : Model
**Output:** $\{\mathbf{a}_1, \mathbf{b}_1, \ldots, \mathbf{a}_k, \mathbf{b}_k\}$ : Convo
1: $\tilde{\mathbf{c}} \leftarrow \texttt{chunk}(\mathcal{C})$                              ▷ **(1)** Get a **subcorpus** of $\mathcal{C}$ that fits in the context window
2: $\mathbf{s} \leftarrow \texttt{get\_seed\_prompt}()$                       ▷ **(2)** Get a prompt to **seed** the first message from $A$
3: **for** $i = 1$ to $k$ **do**                                   ▷ **(3)** Sample a **conversation** with $k$ back and forths
4:     $\mathbf{a}_i \sim \mathcal{F}(\cdot \mid \tilde{\mathbf{c}} \oplus \mathbf{s} \oplus \mathbf{a}_1 \oplus \cdots \oplus \mathbf{b}_{i-1})$   ▷ **(3.1)** Sample $A$'s message with $\tilde{\mathbf{c}}$ and $\mathbf{s}$ in context
5:     $\mathbf{b}_i \sim \mathcal{F}(\cdot \mid \tilde{\mathbf{c}} \oplus \mathbf{a}_1 \oplus \cdots \oplus \mathbf{b}_{i-1} \oplus \mathbf{a}_i)$      ▷ **(3.2)** Sample $B$'s message with $\tilde{\mathbf{c}}$ in context
6: **end for**
7: **return** $\{\mathbf{a}_1, \mathbf{b}_1, \ldots, \mathbf{a}_k, \mathbf{b}_k\}$

---

The conversation is generated by iteratively sampling generations from two LLM participants $A$ and $B$ (which are the same model). We maintain two different conversation histories: $A$'s starts with a *user* message containing a seed prompt $s$ (*e.g. "Please start a conversation by asking a question about the document above."*) followed by alternating *assistant* and *user* messages from $A$ and $B$, respectively. $B$'s conversation history does not include the seed prompt and contains the same messages as $A$'s but with the roles of $A$ and $B$ swapped. Both have the subcorpus $\tilde{\mathbf{c}}$ in the system prompt. To build a training dataset, we sample $m_{\text{train}}$ independent conversations and concatenate the messages from $A$ and $B$ into a single sequence of tokens:

$$\mathcal{D}_{\text{train}} = \{\mathbf{x}^{(j)} = \mathbf{a}_1^{(j)} \oplus \mathbf{b}_1^{(j)} \oplus \mathbf{a}_2^{(j)} \oplus \mathbf{b}_2^{(j)} \oplus \cdots \oplus \mathbf{a}_k^{(j)} \oplus \mathbf{b}_k^{(j)}\}_{j=1}^{m_{\text{train}}} \tag{1}$$

where each $\mathbf{x}^{(j)}$ is a concatenation of the messages. Note that all of the datasets on which we evaluate in the main paper involve a single-turn. So, we set $k = 1$, generating a synthetic conversation with one user message and one assistant message.

Note that the $\texttt{chunk}$ and $\texttt{get\_seed\_prompt}$ functions expose two different ways to control the data distribution of the synthetic data. We find that these two design decisions are critical for training high quality CARTRIDGES with SELF-STUDY.

**Chunking**   We use short subcorpora $\tilde{c}$ (between 512 and 4096) tokens to let the LLM focus on different parts of the corpus when generating data. This is motivated by observations in prior work (Liu

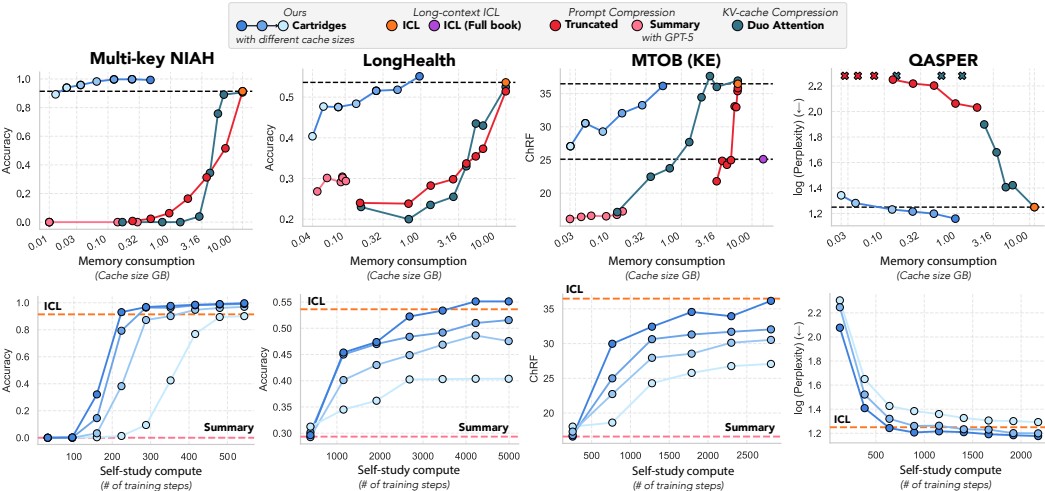

Figure 3: CARTRIDGES **can match ICL quality with lower memory costs by scaling SELF-STUDY compute. (Top)** We measure response quality ($y$-axis) against KV cache memory consumption ($x$-axis) for different methods, at different KV cache sizes. The dashed line marks the quality of standard ICL. **(Bottom)** We measure response quality ($y$-axis) against scale of self-study compute ($x$-axis). The dashed line marks the quality of ICL and prompt compression baselines.

et al., 2024c; Narayan et al., 2025). Furthermore, chunking also allows us to train CARTRIDGES on corpora longer than the model's context window.

**Seed prompts** Instead of using just one seed prompt, we curate a list of five different seed prompt types: *structuring*, *summarization*, *question*, *use cases*, and *creative*. The full list of seed prompts used in our experiments is provided in Section C. Critically, in all our experiments the seed prompts are **generic**: they do not mention anything related to the specifics of the corpora we evaluated (*e.g.* no mention of translation for MTOB or medical terms for LongHealth). We use the same set of seed prompts across all of the experiments. In Section 5.3, we ablate the use of diverse seed prompts and find that it improves performance over a single generic seed prompt by up to $4.8$ accuracy points ($43.6 \rightarrow 48.4$ on LONGHEALTH).

## 4.2 SELF-STUDY CONTEXT-DISTILLATION OBJECTIVE

Given a fine-tuning dataset $\mathcal{D}_{\text{train}}$, we adapt standard techniques from the model distillation literature (Kim & Rush, 2016; Snell et al., 2022; Kujanpää et al., 2024). We let $\mathcal{F}(\cdot|\mathbf{x})$ denote the next token distribution given some input text $\mathbf{x}$. Our *teacher* is the model with the subcorpus, $\tilde{\mathbf{c}}$, in context $\mathcal{F}(\cdot|\tilde{\mathbf{c}})$ and our *student* is the same model adapted with a trainable cache $\mathcal{F}_Z(\cdot)$. We use a classic distillation objective (Hinton et al., 2015) that minimizes the KL-divergence between the teacher and student next-token distributions over a sequence of tokens $\mathbf{x}$ and the corresponding subcorpus used to generate them $\tilde{\mathbf{c}}$.

$$\arg\min_{Z} \sum_{(\mathbf{x}, \tilde{\mathbf{c}}) \in \mathcal{D}_{\text{train}}} \sum_{i=1}^{|\mathbf{x}|} D_{\text{KL}}\bigg( \mathcal{F}(\cdot|\tilde{\mathbf{c}} \oplus \mathbf{x}[:i]) \quad || \quad \mathcal{F}_Z(\cdot|\mathbf{x}[:i])\bigg) \tag{2}$$

In Section A, ablate the use of the context-distillation objective and show that improves accuracy when controlling for the amount of synthetic data (*e.g.* $3.7$ accuracy points on LONGHEALTH).

## 5 RESULTS

We describe experiments evaluating the effectiveness of CARTRIDGES trained with SELF-STUDY in various long-context scenarios. Our results support the following claims. **First**, CARTRIDGES trained with SELF-STUDY can match or outperform ICL while maintaining generality and reducing serving costs (Section 5.1). **Second**, SELF-STUDY is effective on corpora longer than the context window of the LLM (Section 5.2). **Third**, the parameterization ablations to assess the relative benefits of

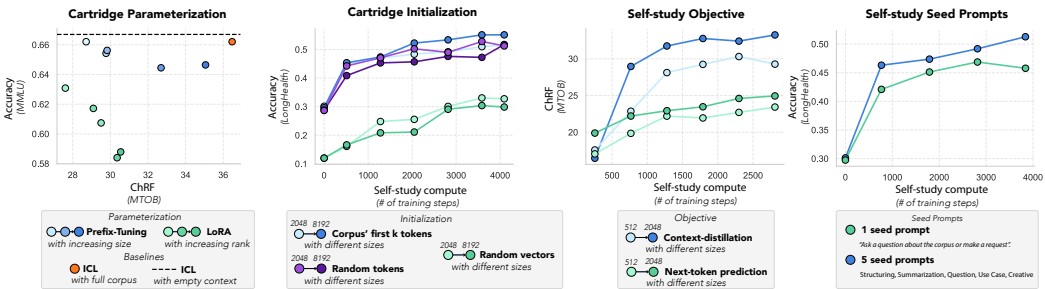

Figure 4: **Ablating CARTRIDGE and SELF-STUDY design choices.** Here we include ablations for parameterization, initialization, objective, and seed prompts on the MTOB or LONGHEALTH datasets (see Section A for full ablation experiments on all datasets).

different aspects of SELF-STUDY and CARTRIDGES (Section 5.3). **Fourth**, when we concatenate two different CARTRIDGES without any joint training, the model can respond to queries requiring information from both CARTRIDGES (Section 5.4).

**Datasets** We study datasets consisting of diverse $(q, r)$ pairs about a single long document. Across datasets, $\mathcal{C}$ ranges between 100k and 484k tokens. Our datasets are drawn from popular long-context benchmarks, with some used as-released and others modified to meet this structure. These include: Multi-key Needle-in-a-Haystack (NIAH) (Hsieh et al., 2024), LONGHEALTH (Adams et al., 2024), MTOB (Tanzer et al., 2023), and QASPER (Dasigi et al., 2021). We evaluate LLM response quality using accuracy for NIAH and LONGHEALTH, log perplexity for QASPER, and character n-gram f-score (chrF) for MTOB (Tanzer et al., 2023; Popović, 2015). Because each dataset effectively consists of a "single" document, we train a single CARTRIDGE per dataset and evaluate it on the queries response pairs $(q, r)$. Section D provides further details.

## 5.1 EXPANDING THE QUALITY-MEMORY FRONTIER BY SCALING SELF-STUDY COMPUTE

We assess how CARTRIDGES produced with SELF-STUDY fare in quality and memory consumption against baselines on the NIAH, LONGHEALTH and QASPER datasets. For all three datasets, $\mathcal{C}$ fits within the model context window (128k tokens). We compare to traditional ICL, two prompt compression baselines (prompt truncation and prompt summarization using GPT-4o (OpenAI, 2024)), and the state-of-the-art KV cache compression baseline ((Jiang et al., 2023a; Xiao et al., 2024b)). Please see Section A.1 for comparisons with other cache compression baselines. We evaluate memory use in terms of KV cache size: the size of the KV cache for the ICL model and prompt compression methods, the size of the CARTRIDGE, and the size of the compressed KV cache for KV cache compression methods like DuoAttention.

The top of Figure 3 presents our main results on LLAMA 3. Compared with ICL, CARTRIDGES offers substantial memory savings at comparable performance: up to $13.8\times$ smaller for LONGHEALTH, up to $97.0\times$ for QASPER, and up to $648.3\times$ for NIAH. As Figure 2 (right) shows, these memory reductions translate to peak throughput (tokens/s) increases of $11.5\times$ and $76.6\times$ for LONGHEALTH and QASPER, respectively. In contrast, all of the cache compression baseline methods fail to match ICL quality even at modest compression ratios of $2-4\times$. See Section A.2 for results with the QWEN3 family of models, where we observe even larger compression ratios: CARTRIDGES $106.4\times$ smaller outperform full ICL KV caches by 3.8 accuracy points on LONGHEALTH.

These substantial compression ratios are not a free lunch. As we show in the bottom of Figure 3, achieving ICL quality at large compression ratios requires spending between two to four orders of magnitude more compute (FLOPs) than we would running prefill with standard ICL. The value of SELF-STUDY, is that it gives practitioners the option to trade off increased offline compute for reduced online memory consumption, which is advantageous in settings where users care about time-to-first-token and latency, users issue many queries over the same corpus, or when we have access to cheap offline compute resources (*e.g.* at night when user load is low (Jaiswal et al., 2025; Goel et al., 2025)). Notably, on NIAH, LONGHEALTH, and QASPER, we observe that when we scale compute, performance improves steadily and eventually exceeds ICL quality.

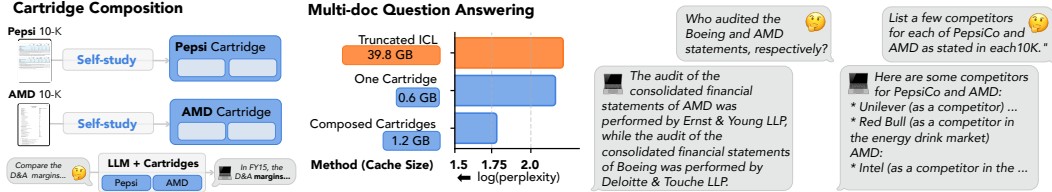

Figure 5: **CARTRIDGE Composition.** (**Left**) Illustration of CARTRIDGE composition, where two independently trained CARTRIDGES (one for a Pepsi 10-K and one for an AMD 10-K) are concatenated without any additional training. (**Middle**) We evaluate composition on a dataset of multi-document questions requiring information in two different $\approx$100k token documents with LLAMA-3B (see Section D). The $x$-axis shows log-perplexity (lower is better) on gold-standard answers. We compare CARTRIDGE composition with an (a) ICL baseline where we truncate the document to fit in the 128k token context length and (b) an CARTRIDGE baseline where we only include the CARTRIDGE for one of the documents. (**Right**) Examples of responses to multi-document questions using composed cartridges.

## 5.2 EXTENDING THE EFFECTIVE CONTEXT WINDOW WITH SELF-STUDY

We evaluate whether SELF-STUDY allows us to accurately process corpora that exceed the context window length. To study this, we consider the MTOB dataset, and LLAMA-8B, which has a context window of 128k tokens. MTOB provides two different long documents: a full 484k token latex textbook and a shorter 60k token version, which was manually-curated by the dataset authors to exclude content not relevant to the translation task. Even though the 484k textbook is 356k tokens *longer* than LLAMA-8B's context window length, we can produce a CARTRIDGE for the full textbook using the chunking strategy of SELF-STUDY. Figure 3 (middle plot) shows the performance of CARTRIDGES of various sizes trained with SELF-STUDY.

As a point of comparison, we provide the results for KV cache baseline methods on the smaller 60k token textbook, and also include ICL on a truncated version of the long textbook. Like above, we observe that CARTRIDGE can match the performance of ICL on the hand-curated 60k token version, while requiring substantially less memory and only having access to the 484k token version, which exceeds the context window of LLAMA-8B. CARTRIDGES also outperform competitive baselines at every KV cache size, by up to 11.0 chrF points.

## 5.3 ABLATING SELF-STUDY DESIGN CHOICES

We perform ablations to study different aspects of SELF-STUDY and CARTRIDGE parameterization, with full results in Appendix A and key findings highlighted in Figure 4.

First, we ablate the parameterization and initialization of CARTRIDGES. We find that the prefix-tuning parameterization substantially outperforms LoRA: on MTOB with CARTRIDGES $\approx$ 0.6 GB, prefix-tuning achieves 4.5 ChRF points higher performance. More importantly, prefix-tuning maintains generalization to unrelated queries (MMLU accuracy drops only from 54.7 to 54.3 as CARTRIDGE size increases from 0.15 GB to 0.96 GB), while LoRA suffers severe degradation (from 54.7 to 45.3 accuracy). Initializing the CARTRIDGE with the KV cache of the first $p$ tokens of the corpus achieves 55.3% accuracy on LONGHEALTH compared to only 29.9% with random vectors. Interestingly, simply initializing with the KV cache of a different corpus closes most of the gap, achieving 51.3% accuracy. See Figure 6 and Figure 8 for complete results on other datasets.

Next, we ablate SELF-STUDY design choices. We find that context-distillation objective significantly outperforms standard next-token prediction, improving ChRF by 8.6 points on MTOB ($24.9 \rightarrow 33.5$) with similar gains on LONGHEALTH and QASPER. Further, we show that using a diverse set of five generic seed prompts (provided verbatim in Section C.1) improves performance over a single prompt (*"Please generate a single chat message to begin a conversation about the information in the corpus. Ask a question about the corpus or make a request."*): +7.9 ChRF points on MTOB ($24.1 \rightarrow 32.0$) and +4.8 accuracy points on LONGHEALTH ($43.6 \rightarrow 48.4$).

## 5.4 COMPOSING CARTRIDGES

We evaluate if independently trained CARTRIDGES can be *composed* (*i.e.* concatenated along the sequence dimension) in order to serve queries about two different corpora (see Figure 5). We train CARTRIDGES across sizes $\{512, 1024, 2048, 4096\}$ and long 10-K documents from AMD, Pepsi, AMEX, and Boeing (Islam et al., 2023). For each pair of CARTRIDGES pairwise (6 pairs per cache size), we evaluate using a dataset of *multi-document questions*, i.e., requiring information from both 10-Ks. Surprisingly, we find composition not only leads to coherent LLM generations *off-the-shelf without any re-training* (Figure 5), but also substantially outperforms the use of a single CARTRIDGE (*i.e.* for only AMD) or ICL (which struggles due to context length limits) (Figure 5) on the multi-document questions.

## 6 DISCUSSION AND CONCLUSION

We propose CARTRIDGES as an alternative to ICL for settings where many different user messages reference the same large corpus of text.

There are several limitations of this work. **First**, this work does not strive to reduce the SELF-STUDY training cost and there is ample room for future optimizations that would make SELF-STUDY training procedure less costly (*e.g.* shared-prefix attention kernels (Ye et al., 2025) or improved synthetic data mixtures (Chen et al., 2024b)). **Second**, in our work, CARTRIDGES matches ICL quality on the LongHealth benchmark, which tests long-distance dependencies, and on MTOB, which is cumulative. However, there remains headroom on these benchmarks and other domains with long-distance dependencies (e.g. code repositories). Future work should explore improvements to self-study that would enable it to better handle cumulative corpora and long-term dependencies. **Third**, in this work, we share the surprising result that when we concatenate two different CARTRIDGES without any joint training, the model can respond to queries requiring information from both CARTRIDGES. However, we stop short of the stronger claim that CARTRIDGES are as effective when composed as they are when used in isolation. Future work should explore how to more effectively compose CARTRIDGES.

This work demonstrates that it is possible to trade off increased offline compute for reduced KV cache memory consumption. Looking forward, this could pave the way to new context-aware AI applications that are currently bottlenecked by memory consumption, from medical assistants that know a patient's full medical history to LLM-powered IDEs that understand entire codebases.

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

**Note on LLM usage** We used LLMs for polishing or improving the grammatical correctness of the writing in this paper. We also used LLMs to identify related work and write code.

# A EXTENDED RESULTS

In this section, we include additional cache compression baselines, report results on an additional model family, and ablate the main design choices of CARTRIDGES and SELF-STUDY.

## A.1 COMPARISON WITH ADDITIONAL CACHE COMPRESSION BASELINES

| | MTOB | | | Longhealth | | |
|---|---|---|---|---|---|---|
| **Method** | ChRF | # cache tok. | Compression | Accuracy | # cache tok. | Compression |
| Full ICL | **36.5** | 48k | 1× | **53.6%** | 114k | 1× |
| CARTRIDGE | 30.5 | 256 | **188×** | 47.7% | 512 | **223×** |
| AdaKV | 29.1 | 9.6k | 5× | 41.8% | 23k | 5× |
| KeyDiff | 27.1 | 9.6k | 5× | 40% | 23k | 5× |
| TOVA | 24.7 | 9.6k | 5× | 32.2% | 23k | 5× |
| SnapKV | 29.7 | 9.6k | 5× | 33.1% | 23k | 5× |

Table 1: Comparison of CARTRIDGES, ICL baseline, and additional cache compression baselines on MTOB and LongHealth.

In Figure 3, we include comparisons with additional cache compression baselines a very strong GPT-4o based summary prompt compression method and Duo-attention (the strongest cache compression method in NVidia's KVPress library (Jegou et al., 2024)). Here, we include results for the next four best performing cache compression methods

## A.2 EXPERIMENTS WITH THE QWEN3 FAMILY OF MODELS

| | MTOB | | | Longhealth | | |
|---|---|---|---|---|---|---|
| Method | ChRF | # cache tok. | Compression | Accuracy | # cache tok. | Compression |
| Full ICL | 25.8 | 48k | 1× | 51.2% | 109k | 1× |
| CARTRIDGE | 32.43 | 4096 | 11.7× | **56.0%** | 4096 | 26.6× |
| CARTRIDGE | **33.27** | 2048 | 23.4× | 55.5% | 2048 | 53.2× |
| CARTRIDGE | 32.3 | 1024 | 46.9× | 54.0% | 1024 | 106.4× |

Table 2: Performance of QWEN3 4B CARTRIDGES on MTOB and Longhealth with various sizes $p$.

In Figure 3, we report results for the Llama-3 family of models. To confirm that our results are not specific to that one family of models, we also report results for the Qwen3 family of models in this section. With Llama on the LongHealth we were able to achieve equivalent quality to ICL with 10x smaller caches, on average. With Qwen the compression ratio is even larger: on longhealth, we outperform the full KV cache by 3.8 accuracy points while being 106.4x smaller. The results are presented in Table 2.

## A.3 CARTRIDGE DESIGN CHOICES: PARAMETERIZATION AND INITIALIZATION

In our experiments, we parameterize the CARTRIDGE with a simplified version of prefix-tuning and initialize with a truncated KV-cache (see Section 3.2). In this section, we describe ablation experiments motivating these design choices. First, we compare two different CARTRIDGE parameterizations (Figure 6): simplified prefix-tuning (Li & Liang, 2021) and low-rank adaptation (LoRA) (Hu et al., 2022). Then, we demonstrate the importance of proper CARTRIDGE initialization (Figure 8).

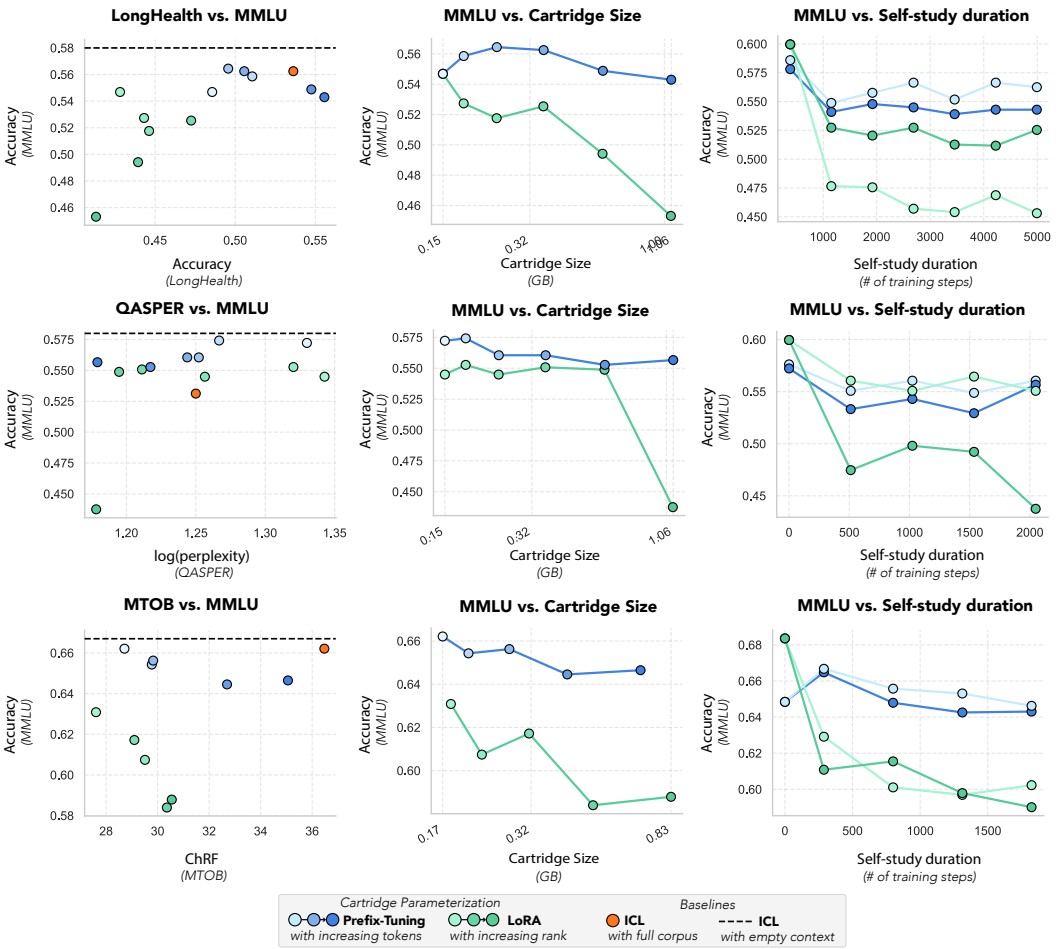

Figure 6: **Comparing CARTRIDGE parameterizations.** We train CARTRIDGES using SELF-STUDY on the corpora from LONGHEALTH (Top), QASPER (Middle), and MTOB (Bottom) using two different parameterizations: simplified prefix-tuning (as described in Section 3.2) and low-rank adaptation (LoRA) (Hu et al., 2022). We experiment with different CARTRIDGE sizes and choose LoRA rank and prefix-tuning cache size to align on memory consumption. We evaluate the performance of the CARTRIDGES on questions from the target dataset (LONGHEALTH or QASPER) using the same protocol as in Figure 3 and also on questions from MMLU (Hendrycks et al., 2020) that are unrelated to the corpora. (**Left**) The $x$-axis shows accuracy on MMLU and the $y$-axis shows accuracy on the target dataset. Each point represents a different CARTRIDGE size. (**Center**) The $x$-axis shows CARTRIDGE size in GB, and the $y$-axis shows accuracy on MMLU. (**Right**) The $x$-axis shows self-study duration in training steps, and the $y$-axis shows accuracy on MMLU. The shade of the points represents the size of the CARTRIDGE.

| Method | Consumes limited memory | Retains corpus information | Supports diverse prompts |
|---|:---:|:---:|:---:|
| In-context learning | ✗ | ✓ | ✓ |
| Prompt / KV cache compression | ✓ | ✗ | ✓ |
| CARTRIDGE + Next-token-prediction | ✓ | ✓ | ✗ |
| CARTRIDGE + SELF-STUDY | ✓ | ✓ | ✓ |

Figure 7: **Comparing KV caching strategies.** CARTRIDGE improves memory efficiency, while retaining the quality of in-context learning across a broad set of prompts. ✓ indicates a strength and ✗ indicates a limitation.

**Parameterization**    We evaluate CARTRIDGES trained on corpora from LONGHEALTH or QASPER on both *in-domain* (*i.e.* questions from LONGHEALTH or QASPER) and *out-of-domain* (*i.e.* questions from an unrelated benchmark, MMLU (Hendrycks et al., 2020)) queries.

We find that the prefix-tuning parameterization is more effective than a memory-matched LoRA parameterization on both in-domain and out-of-domain queries. This is illustrated in Figure 6 (Left), where we see that prefix-tuning occupies the top-right corner of the plot (high accuracy on both MMLU and the target dataset).

Notably, we find that as we increase the CARTRIDGE size with LoRA tuning, performance on out-of-domain queries (MMLU) drops significantly. At 1.06 GB (LoRA rank 1632), MMLU accuracy drops from 60.0% to 45.3%. This drop in performance is highly correlated with the size of the CARTRIDGE, suggesting that LoRA is not well-suited to large Cartridges, which we show in Figure 3 are important for recovering ICL performance. In contrast, with prefix-tuning the accuracy only drops to 54.3% at 1.06 GB. This degradation is mostly invariant to the size of the CARTRIDGE (54.7% at 0.15 GB), demonstrating that out-of-domain performance is robust across CARTRIDGE sizes.

On in-domain queries, prefix-tuning also outperforms LoRA, but the gap is smaller. Across all CARTRIDGE sizes, the best LONGHEALTH accuracy prefix-tuning achieves is 55.6% at 0.96 GB, while the best LoRA accuracy is 47.25% at 0.26 GB. Interestingly, LoRA accuracy at the largest CARTRIDGE sizes is lower; 41.3% at 0.96. It is possible that this is due to the out-of-domain degradation of LoRA we discussed above. Since queries in LONGHEALTH test set are quite different from the synthetic queries generated by SELF-STUDY (*e.g.* they are multiple choice and require some complicated reasoning traces), out-of-domain robustness may be also important for "in-domain" performance.

It isn't clear why prefix-tuning is so much more robust than LoRA to out-of-domain performance degradation. It is surprising given the similarity between a KV-cache and an MLP – both are linear transformations separated by a non-linearity. It is possible that this is due to the difference in the activation function (SiLU vs. Softmax). We leave a more detailed investigation into the root cause of this difference for future work.

**Initialization**    The standard way of initializing a $k$ token CARTRIDGE in our main paper is using the KV cache from the first $k$ tokens of the source document. In Figure 8, we ablate different initialization source. We try two additional initalizations: *random vectors* and *random tokens*.

For *random vectors*, we simply initialize the parameters of the CARTRIDGE from a component-wise standard normal distribution. For *random tokens*, we initialize the CARTRIDGE as the KV cache of the first $k$ tokens of arbitrary text (specifically, the Wikipedia page for gradient). The important difference between the these two strategies is that for *random tokens* the initial CARTRIDGE is "valid" KV cache produced by the model, while for *random vectors* it is not.

**Freezing the attention sink**    A small yet important detail of training a CARTRIDGE is that we do not let the first token's key and value vectors to be trainable. As studied in (Xiao et al., 2024c), the first key vector, which corresponds to the beginning of sequence token and is thus the same for *every sequence*, acts as an "attention sink". We observed that when training a CARTRIDGE, allowing those key and value vectors to be trainable led to training instability (see Figure 9). For example, on some runs the MMLU accuracy would dip to below 30%.

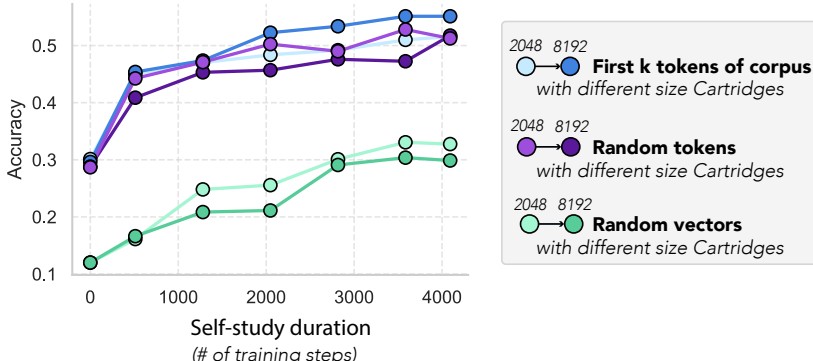

Figure 8: **Ablating CARTRIDGE initalization**. We train a CARTRIDGES using SELF-STUDY on the corpora from LONGHEALTH with 3 different initialization strategies. The $x$ axis is the number of training steps and the $y$ axis is the accuracy on LONGHEALTH. The blue lines are the results when initializing the CARTRIDGE using the KV cache from the first $k$ tokens of the document. The purple lines are initializing the CARTRIDGE from the KV cache of unrelated text. The green lines is initializing the CARTRIDGE with random vectors. Initializing from the first $k$ tokens leads to slightly stronger results than initializing from the KV cache of random text. This difference may be more prominent on other corpora where the first $k$ tokens are more relevant to solving the downstream task.

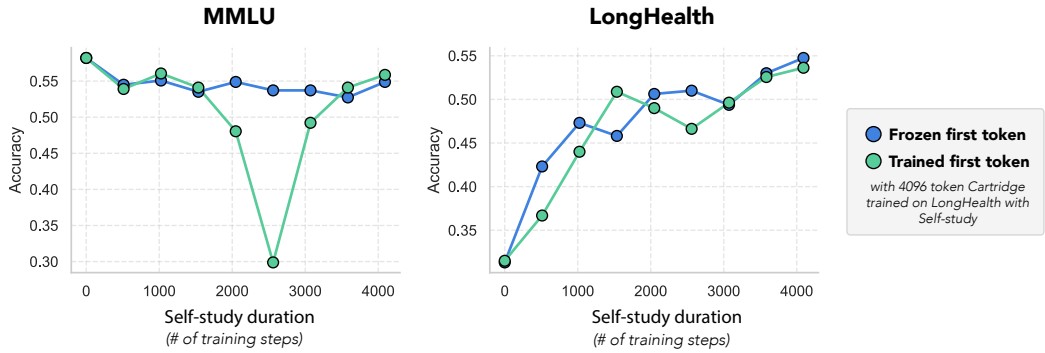

Figure 9: **Freezing the attention sink**. In both plots, the y-axis is accuracy and the x-axis is training step. The green line which corresponds to a run where we allow a trainable first token. (**Left**) The y-axis MMLU accuracy. This plot exemplifies the training instability we observed when the key and value vectors were trainable. The MMLU score dips to below 30% before recovering. (**Left**) The y-axis is accuracy on questions from LONGHEALTH.

## A.4 SELF-STUDY DESIGN CHOICES: DATA-GENERATION AND OBJECTIVE

In SELF-STUDY training we use a seeded data-generation process and a context-distillation training objective (see Section 4). In this section, we ablate these design choices, comparing against the performance of SELF-STUDY with simpler data-generation and objectives.

**Data Generation** In Section 4.1, we describe how we use five different seed prompt types when generating data with Algorithm 1. These prompt types, *structuring*, *summarization*, *question*, *use cases*, and *creative*, are described in more detail in Section C.1.

In this section, we compare the performance of SELF-STUDY with these five prompt types against SELF-STUDY with a single prompt: *"Please generate a single chat message to begin a conversation about the information in the corpus. Ask a question about the corpus or make a request."*

Across three datasets, we find that using the five different prompt types during SELF-STUDY leads to higher quality CARTRIDGES (see Figure 11). On MTOB with CARTRIDGES of size 1024 tokens,

we see a 7.9 point ChRF improvement ($24.1 \rightarrow 32.0$). On LONGHEALTH, the improvement is 5.5 accuracy points ($45.8 \rightarrow 51.3$).

Interestingly, on QASPER, we see no benefit from using the five different prompt types. It is possible this is because the queries in the QASPER dataset are mostly factual questions that do not require complex reasoning like LONGHEALTH and MTOB do.

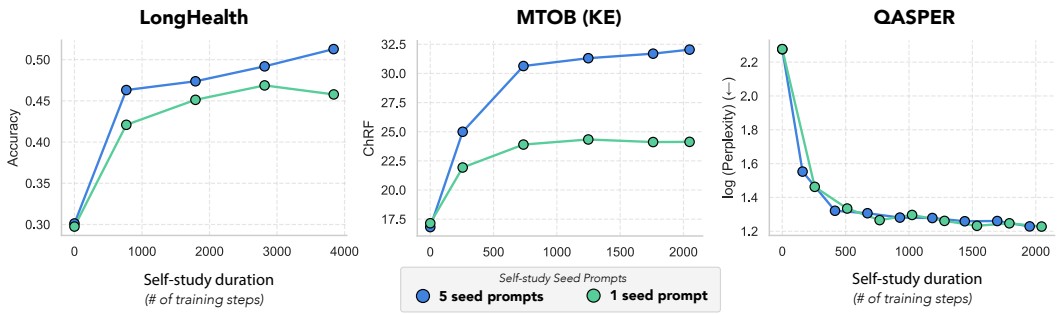

Figure 10: **Diverse seed prompts improve quality.** We generate synthetic data according to Algorithm 1 and ablate the choice of seed prompts sampled on Line 2. We consider two approaches: using a single, broad seed prompt (Green) or randomly sampling one of five different types of seed prompts (Blue). We train CARTRIDGES using self-study with these two strategies on LONGHEALTH, MTOB and QASPER corpora. In all plots, the $x$ axis is the number of training steps, and the $y$ axis is either accuracy (for LONGHEALTH and MTOB) or perplexity on ground truth answer (for QASPER). We use an CARTRIDGE size of 1024 tokens.

**Training Objective** In Section 4, we describe the context-distillation objective we use (Snell et al., 2022; Kim & Rush, 2016; Bhargava et al., 2024). This approach requires that we collect top output probabilities from the in-context model's output distribution during data generation. A simpler alternative would be to just use a next-token prediction objective with a cross-entropy loss.

In our comparison, we find that this simpler objective underperforms the context-distillation objective (see Figure 11). Most notably, on MTOB with 2048 token CARTRIDGES, context-distillation outperforms next-token prediction by 8.3 ChRF points ($24.9 \rightarrow 33.2$). On LongHealth, the gap is 3.7 accuracy points ($47.6 \rightarrow 51.3$).

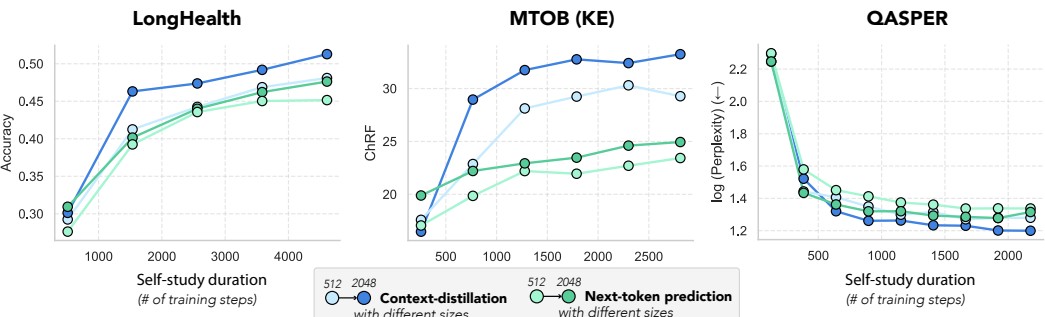

Figure 11: **Context-distillation objective improves training efficiency**. We train CARTRIDGES using SELF-STUDY on the corpora from LONGHEALTH (Left), MTOB (Center) and QASPER (Right) using two loss functions: a next token prediction loss (green) and a distillation loss (blue). We evaluate the performance of the CARTRIDGES on questions from the target dataset (LONGHEALTH, MTOB or QASPER) using the same protocol as in Figure 3. In all plots, the $x$ axis is the number of training steps, and the $y$ axis is either accuracy (for LONGHEALTH and MTOB) or perplexity on ground truth answer (for QASPER). The shade of the points represents the size of the CARTRIDGE. Using a distillation loss achieves higher accuracy (or lower perplexity for QASPER) across datasets and CARTRIDGE sizes.

As shown in Figure 11, quality seems to be consistently improving with more SELF-STUDY compute. It is possible, therefore, that by spending more during SELF-STUDY with the next-token prediction objective, we could close the gap. However, for a fixed amount of SELF-STUDY compute, context-distillation is considerably more effective.

These results demonstrate how context-distillation plays an important role in efficiently recovering ICL performance with SELF-STUDY.

### A.5 Results on synthetic RULER tasks

In this section, we include results on synthetic tasks from the RULER framework (Hsieh et al., 2024). We focus on synthesizing RULER tasks that reflect the intended use case for Cartridges: settings in which many queries reference one context. The three tasks we evaluate are: Multi-Key Multi-Query Single-value NIAH, Multi-Key Multi-Query Single-value NIAH, and Variable Tracking. Note: that for all of our RULER evaluations, we did not change the generic seed prompts from the ones we used in the evaluations from the original paper. We ran all of the experiments below with Llama 3.2 3B at 94k context length.

**Multi-Key Multi-Query Single-value NIAH.** On multi-key, single-value NIAH, Cartridges achieves near perfect accuracy ($> 99\%$), while compressing the context up to $92\times$. Further, we performed a qualitative evaluation of the Cartridges trained on the NIAH corpus (at $92\times$ compression). The model with the Cartridge can summarize the background "haystack" context (a random assortment of essays from the investor Paul Graham):

*User: In a sentence, can you summarize what you learned about?*

*Assistant: The text discusses various topics including the challenges of designing a language that programmers will love, ... and the value of learning from others, including hackers and experts in the field.*

We also checked that the model is capable of extracting the needle (abundant-young=9375247) when prompted in non-standard ways:

*User: What would the output of the following code be: print(magic_numbers["abundant-young"])?*

*Assistant: The output of the code print(magic_numbers["abundant-young"]) would be 9375247.*

**Multi-Key Multi-Query Multi-value NIAH.** The configuration of NIAH, in which the model has to output multiple values per key, is more challenging than the single-value variant above. The full KV-cache achieves only 28.8% exact match accuracy, which Cartridges can match with $23\times$ compression.

Note that the original Ruler paper evaluates only with recall (instead of exact match). We found that the model often had high recall but low precision, so we report exact match to reflect this.

**Variable tracking.** On variable tracking, Cartridges outperforms the full KV cache (26.2% vs. 22.8% accuracy) while being $45.9\times$ smaller.

### A.6 Throughput measurement details

We provide details for the throughput measurements in Figure 2. We use the state-of-the-art SGLang inference system, with default parameters (Zheng et al., 2024). We measure throughput on a single H100 GPU.

We first determine the largest batch size $b$ that fits in GPU memory, given a cache of size $k$ tokens. We then randomly initialize $b$ CARTRIDGES of size $k$ and pre-load the CARTRIDGES into GPU memory. We finally measure the time taken to decode 128 tokens per sequence. The CARTRIDGES and decoded tokens are appended to a KV-cache during generation. We report the average of 5 iterations after using 3 warm-up iterations.

## B EXTENDED RELATED WORK

In this section, we provide a more in-depth discussion of the place our work occupies in the broader literature. The structure below mirrors the structure of our paper: first we discuss work related to the parameterization and initialization of CARTRIDGES (Section B.1), then we cover work that inspired the design of SELF-STUDY (Section B.2), and finally we describe other approaches aimed at reducing the size of the KV-cache, many of which we compare against in our experiments (Section B.3).

### B.1 PRIOR WORK RELATED TO THE PARAMETERIZATION OF CARTRIDGES

Below we discuss prior work from the parameter-efficient fine-tuning literature that inform the way we parameterize CARTRIDGES in our work.

#### B.1.1 PARAMETER-EFFICIENT FINE-TUNING (PEFT)

In order to adapt large language models (LLMs) to particular domains or tasks in a more compute and memory-efficient manner, several parameter-efficient fine-tuning (PEFT) methods have been developed. Some of the most widely used PEFT methods include Low-Rank Adaptation (LoRA) (Hu et al., 2022), prefix-tuning (Li & Liang, 2021), and prompt-tuning (Lester et al., 2021).

Leveraging prior observations that fine-tuned language models exhibit an intrinsic low rank structure, Hu *et al.* propose LoRA, which freezes model parameters and injects trainable rank decomposition matrices between each transformer layer. LoRA exhibits on-par or better fine-tuning quality while reducing the number of trainable parameters by 10,000 times and the GPU memory requirement by 3 times (Hu et al., 2022).

Li *et al.* and Lester *et al.* both take a different approach to lightweight fine-tuning, proposing tunable "prefixes" and "soft prompts" respectively to prepend to queries in order to steer the model to desired outputs. Li *et al.* proposes prefix-tuning, which learns a continuous representation for the activation of the prefix at each transformer layer. These learned activations are then prepended to activations obtained by passing the input prompt through the frozen transformer. In contrast, Lester *et al.* proposes prompt-tuning, which optimizes at the discrete token level and prepends a series of learnable tokens to the input prompt. Both methods show strong performance while greatly reducing the number of learnable parameters and improving compute and memory efficiency for language model adaptation.

Principal Singular values and Singular vectors Adaptation (PiSSA) (Meng et al., 2024) is another more recent PEFT method that attempts to ameliorate the slow convergence problems of LoRA. PiSSA initializes the LoRA rank decomposition matrices with the principal components of the original matrix, and exhibits faster convergence and enhanced performance compared to LoRA on several tasks, including GSM8K and MATH.

Several of these methods, especially LoRA, have been adapted specifically for distilling knowledge provided in context into the parameters of a language model. Some of those methods are described in the sections below, and this work is an extension of prefix-tuning for long-context tasks.

#### B.1.2 PARAMETER-EFFICIENT ADAPTER COMPOSITION AND MERGING

A number of works have explored the idea of composing multiple different parameter-efficient adapters (*e.g.* LoRAs) by summing them together, concatenating them, or using a dynamic mixture of experts (Zhao et al., 2024b; Huang et al., 2023; Xiao et al., 2024a; Zhao et al., 2024a; Yadav et al., 2024; Wu et al., 2024; Gou et al., 2023; Li et al., 2024a). For example, Huang *et al.* propose LoraHub, a framework for dynamically weighting and composing multiple language model adapters (Huang et al., 2023). Given a set of LoRA modules for different upstream tasks and new unseen task with in-context examples, LoraHub dynamically weights the LoRAs and composes a new LoRA module for the task. Similarly, Zhao *et al.* propose a method for dynamically *retrieving* the most relevant language model LoRAs for a given task (Zhao et al., 2024a).

### B.1.3 Parametric Knowledge Injection

Several recent works have explored methods for integrating external knowledge directly into model parameters, known as parametric knowledge injection (Kujanpää et al., 2024; Mao et al., 2025; Su et al., 2025; Caccia et al., 2025; Kuratov et al., 2025). To the best of our knowledge, these studies are the closest in scope to ours. Like ours, these works address the problem of parametric knowledge injection: how to store large text corpora within parameters of a language model. Some use simple synthetic data generation pipelines or context-distillation objectives. Unlike our work, these studies do not highlight the memory reduction and throughput advantages of parametric knowledge injection techniques. We highlight other differences below.

One parametric knowledge injection method, recently proposed by Kujanpaa *et al.*, is prompt distillation, in which a teacher model with access to privileged knowledge generates question-answer pairs. These pairs are then used to train a LoRA adapter for a student model (identical to the teacher model, but without access to privileged information) using a distillation objective (i.e. mimicking the teacher's full token distribution) (Kujanpää et al., 2024). This closely resembles our context-distillation objective, which we also found works better than next-token prediction. However, unlike our work, Kujanpaa *et al.* only train LoRA adapters of a single size (rank 1024) and don't assess memory reductions with respect to full in-context learning. Indeed, they do not evaluate against long-context ICL baselines at all, focusing instead on a comparison with RAG. Furthermore, they evaluate on a relatively simple long-context setting – a concatenation of SQUAD passages (Rajpurkar et al., 2016) – which does not exhibit long range dependencies or require reasoning the way MTOB and LONGHEALTH do.

Similarly, Mao *et al.* propose Long Input Fine-tuning (LIFT), which fine-tunes a language model using a typical next-token prediction objective on overlapping segments of the corpus, as well as instruction tuning on question answer pairs generated from the corpus. Unlike our work, Mao *et al.* find that synthetic Q/A pairs "offer minimal benefit and can even degrade performance due to overfitting" (Mao et al., 2025). The difference in our findings is perhaps due to the fact that they only generate *ten* synthetic examples, whereas we generate *tens of thousands*. Furthermore, they use a weaker ICL baseline (Llama 3 8B) that only has 8k tokens of context. Any contexts longer than 8k tokens are truncated before being fed to the ICL baseline.

Concurrent work on *deep context distillation* performs knowledge injection with synthetic data and a context distillation objective (Caccia et al., 2025). In this work, the authors only report performance with LoRA adapters and do not explore a prefix-tuning parameterization. In further contrast to our work, their focus is not on memory reductions or throughput improvements. They only report performance with a single adapter size (rank 16 LoRA adapters), and they do not report throughput improvements. Instead, the paper highlights the "plug-and-play" nature of the method.

Finally, Su *et al.* proposes Parametric Retrieval Augmented Generation (Parametric RAG), in which each document has a corresponding LoRA adapter, trained on an augmented dataset consisting of the document, rewritten versions of the document, and question-answer pairs generated from the document. At inference time, a retriever is used to determine relevants documents, and the corresponding LoRA adapters are merged (Su et al., 2025). This method demonstrates significant gains over RAG on a variety of tasks, including WikiMultihopQA.

## B.2 Prior work related to Self-Study

### B.2.1 Self Distillation and Context Distillation

Self-distillation is another method used to internalize the performance gains provided by information in context (e.g. scratchpads, informative instructions) into the model parameters. In "Learning by Distilling Context", the authors distill a model with instructions and scratchpads in context into parameters by conditioning the model on "[instructions] + [task-input]" to predict "[scratch-pad] + [final answer]"; then fine-tuning the same model to predict its own "[final answer]" conditioned on the "[task-input]", without seeing the "[instructions]" or using the "[scratch-pad]" (Snell et al., 2024).

### B.2.2 Synthetic Data Generation

Due to the ubiquitous need for high quality data for fine-tuning (e.g. for use with the methods described above), a large body of work has focused on generating high quality synthetic data (Nayak et al., 2024) (Abdin et al., 2024) (Gandhi et al., 2024) (Riaz et al., 2025). For example, Bonito is a model that is fine-tuned to generate synthetic data (Nayak et al., 2024), and MetaSynth is a method proposed by Riaz *et al.* that uses a language model to orchestrate several expert LLMs for domain-specific synthetic data generation (Riaz et al., 2025). The training process for Phi-4, a 14 billion parameter language model, also incorporates significant amounts of synthetically generated data (Abdin et al., 2024). Incorporating synthetic data, in conjunction with new post-training techniques, allows Phi-4 to surpass its teacher model on STEM QA tasks, as well as perform well for its size on reasoning benchmarks. These works demonstrate the potential for synthetic data generation methods to augment the capabilities of language models.

Contemporaneous work by Lin *et al.* proposes a synthetic data generation recipe called Active Reading, which closely resembles self-study (Lin et al., 2025).

### B.3 Reducing the size of the KV cache

In this section, we discuss existing approaches for reducing the size of the KV cache.

First, in Section B.3.3, we describe works that propose architectural changes to the multi-head attention operation, which reduce the memory footprint of the KV cache. Next, in Section B.3.1, we discuss *prompt compression* methods, which reduce the size of the KV cache by converting a long sequence of input embeddings into a shorter one. They can be split into hard-token methods, which output discrete tokens from the vocabulary, and soft-token methods, which output new token embeddings not from the vocabulary. Finally, in Section B.3.2, we describe *KV cache compression* methods. These methods directly modify the key and value matrices in the KV cache. Compared with prompt compression methods, these are more expressive because they can produce a KV cache that no sequence of input embeddings could have produced.

The methodology proposed in our work relies on cache-tuning, which could be viewed as a form of KV cache compression.

### B.3.1 Prompt compression

**Hard-token prompt compression**   Some works aim to reduce the size of KV cache by converting a longer text into a shorter text (Jiang et al., 2023b; Li, 2023; Chuang et al., 2024; Zhang et al., 2024b; Pan et al., 2024). These methods are typically referred to as *hard-token* prompt compression methods because the resulting KV cache comes from discrete tokens from the vocabulary. Compared with soft-token prompt methods, these methods work well with black-box API models.

These methods can be broadly classified into two categories: filtering and summarization based methods. Filtering methods cut text from the original prompt using heuristics such as self-information. For example, LLMLingua and Selective-Context use a smaller LLM to filter a long prompt (*e.g.* dropping redundant tokens) before passing it to the main model (Jiang et al., 2023b; Li, 2023). Summarization methods paraphrase a long prompt into a smaller number of tokens (Chuang et al., 2024).

**Soft-token prompt compression with adapted LLMs**   In one line of work, researchers train a model (typically an adapted LLM) to compress a long prompt into a smaller number of soft tokens (Chevalier et al., 2023; Yen, 2024; Ge et al., 2023b; Mu et al., 2023; Qin et al., 2023).

For example, *Autocompressors* and *In-context Autoencoders* (ICAE) are LLMs that are fine-tuned to output embeddings which can be used in soft-token prompts (Chevalier et al., 2023; Ge et al., 2023b). Autocompressors are trained with full-parameter fine-tuning and leverage a recursive strategy to generate the soft prompts, whereas ICAEs are trained with LoRA and use a single forward pass to generate the soft prompts. A recent method, LLoCO, train domain-specific LoRA adapters that enable the decoder better leverage AutoCompressor embeddings (Tan et al., 2024). This differs from CARTRIDGES in that the LLoCO LoRA adapters are trained for a domain (*e.g.* academic papers, news), not a specific document. A number of other works also propose using an auxiliary model

to produce soft-tokens from a long prompt (Ge et al., 2023b; Qin et al., 2023). *Gisting* is another method that differs from those above in that it uses the same LLM to compress the prompt into soft tokens as it uses to generate the response (Mu et al., 2023).

**Soft-token prompt compression via gradient-descent**   Soft tokens can also be produced by optimizing input token embeddings with gradient descent. This idea, called *prompt tuning*, was first proposed for the purpose of conditioning a frozen langauge model to perform specific tasks (Lester et al., 2021). As such, it is an important part of the parameter-efficient fine-tuning literature and is discussed in more detail in Section B.1.1. Since then, Li *et al.* has extended prefix tuning techniques to long-context settings, proposing a new method called prefix propagation, which conditions prefixes on previous hidden states to achieve superior performance on long-document tasks compared to prefix tuning (Li et al., 2024a).

### B.3.2    KV CACHE COMPRESSION

**Hard-token KV cache compression**   Motivated by the observation that, in some settings, a small number of keys dominate the attention scores of subsequent queries, several works have proposed *KV cache eviction policies* wherein keys and values are dynamically dropped during generation (Ge et al., 2023a; Zhang et al., 2023b; Tang et al., 2024; Oren et al., 2024). For example, H20 drops keys and values from *generated tokens* based on a running sum of historical attention scores (Zhang et al., 2023b). Similarly, SnapKV drops keys and values from *prompt tokens* based on a window of queries from the end of the prompt (Li et al., 2024b).

A major limitation of eviction methods is that once a key is evicted, it cannot be recovered. Instead of evicting keys permanently, another line of work focuses on selectively loading keys from KV cache to SMs. While these works do not reduce memory consumption of the KV cache, they can speed up inference by making better use of GPU memory bandwidth (Ribar et al., 2023; Tang et al., 2024). For example, the Quest method estimates critical tokens at each decoding step and selectively loads them to SMs (Tang et al., 2024).

Compared with the hard-token *prompt compression* methods, KV-cache compression methods allow fine-grained control at the level of an attention head. This means that a token can be dropped from one attention head but not another.

**Soft-token KV cache compression with merging**   In another line of work, instead of evicting tokens from the KV cache, researchers propose merging similar tokens  (Wang et al., 2024; Zhang et al., 2024d; Wan et al., 2024; Liu et al., 2024b). For example, Cache Merge (CaM) takes keys marked for eviction and merges them instead, using a weighting scheme based on attention weights (Zhang et al., 2024d). Wang *et al.* builds on this work by clustering key states into "merge sets" based on cosine similarity, and merging states within a "merge set" with a Gaussian kernel weighting scheme, which upweights states more similar to a pivotal state chosen as the token with the largest total attention score (Wang et al., 2024). Wan *et al.* expands on both these works with Dynamic Discriminative Operations (D2O), which performs optimizations at both the layer and token levels. D2O adjusts the KV cache budget for each layer based on its attention density and uses an exponential moving average mechanism to dynamically determine when a previously discarded token is similar enough to retained tokens to be merged back in (Wan et al., 2024). All of these works demonstrate promising results, offering similar or better performance on several tasks compared to a full cache with a 50% or more reduction in cache size. However, there is still room for further improvement, as these methods still fail to match full cache performance in several tasks, and even a 50% reduction in cache size may still be prohibitively expensive for very large models or very long contexts. Additionally, these works do not evaluate the effectiveness of these methods in long-context settings.

**Soft-token KV cache compression with low-rank projection**   A number of works leverage the observation that the KV cache exhibits low-rank structure to develop compression methods  (Yu et al., 2024; Chang et al., 2024; Zhang et al., 2024c; Zhou et al., 2025; Saxena et al., 2024). Similar to compression methods based on merging, compression methods based on low-rank adaptation achieve performances similar to or exceeding full caches on several tasks at 50% compression, while experiencing performance degradation upon further compression.

**Soft-token KV cache compression with adapted LLMs**    Above we discussed how some works adapt an LLM to output a shorter sequence of soft tokens given a long context. Similarly, one could adapt an LLM to output a smaller KV cache given a long context. While less explored than the analagous prompt compression approach, there is at least one published method that falls into this category. In *KV-distill*, the authors add LoRA adapters to an LLM's query projections and train them to to produce queries which aggregate information from prior tokens (Chari et al., 2025). The adapter is applied selectively to some tokens and only these tokens are kept in the KV cache. The idea is that these selected tokens can act as sinks to collect information from prior tokens. The adapter is trained with a distillation objective between a compressed and uncompressed KV cache. However, unlike our work, KV-distill does not use any training at test time.

**Soft-token KV cache compression with gradient-descent**    The idea of treating the keys and value matrices in a KV cache as weights and training them with gradient descent was first discussed in the prefix-tuning paper (Li & Liang, 2021). In this work, the method was not applied to long-contexts, but rather as a parameter-efficient fine-tuning method that can be applied to training datasets with input-output pairs, so we discuss it in more detail in B.1.1. Since then, we are not aware of works that have applied this technique to handle long-contexts.

### B.3.3    ARCHITECTURAL CHANGES

A number of works have proposed architectural changes to the original multi-head attention (MHA) operation (Vaswani et al., 2017) that reduce the memory footprint of the KV cache. Because they fundamentally alter the architecture, these methods are not immediately compatible with pre-trained models using the standard MHA operation.

The earliest works in this direction developed fixed sparsity patterns in the attention map (Beltagy et al., 2020; Child et al., 2019; Zaheer et al., 2020). For example, many works use a sliding window sparsity pattern wherein each token attends to a fixed window of tokens around it. These approaches reduce the size of the KV cache because they require only keeping around a fixed number of tokens in the KV cache. More recently, some large language models have adopted sliding window sparsity in a subset of layers/heads (Team et al., 2024).

While the methods above reduce the size of the cache by introducing sparsity at the token-level, another class of methods changes the structure of the attention heads. Multi-query attention (MQA), the earliest of such modifications, uses multiple query heads but only a single key and value head (Shazeer, 2019). While MQA dramatically reduces the size of the KV cache, it can lead to a significant drop in the expressive power of the model. Grouped-query attention (GQA) is a middle ground between MQA and MHA that allows a group of query heads to attend to a single key and value head (Ainslie et al., 2023). Many frontier models use GQA, including the Llama 3 architecture, which we use in our experiments (Dubey et al., 2024; Jiang, 2024; Yang et al., 2024a). More recently, a number of other architectural modifications have been proposed including including Multi-head Latent Attention (Liu et al., 2024a) and Tensor Product Attention (Zhang et al., 2025).

In another line of work, researchers observe that without the softmax operation in the attention mechanism (*i.e.* linearizing the attention operator), the KV cache can be faithfully represented by the fixed size matrix $K^\top V$ (Arora et al., 2024). This allows us to represent the KV cache with a single matrix whose size is independent of the context length.

Indeed, a large body of work has focused on developing architectures with fixed-size memory consumption (*i.e.* models that do away with the KV cache). Notable examples include state-space models (Gu & Dao, 2023), RNNs (Beck et al., 2024), and other linear attention variants (Arora et al., 2024; Yang et al., 2024b).

Prior work shows that there are tradeoffs between the memory consumption of an architecture and the ability of a model to perform recall-intensive tasks, when controlling for compute (*i.e.* FLOPs) (Arora et al., 2024). In this context, our work shows that by increasing compute (*i.e.* FLOPs), we can reduce the memory consumption of a model without sacrificing performance. In Section E, we provide a prelinary theoretical analysis relating SELF-STUDY with recurrent architectures. However, future work should explore the relationship between CARTRIDGES and recurrent models in more depth.

Most related to our work are recent architectures (*e.g.* Titans (Behrouz et al., 2024), TTT (Sun et al., 2024)) that use a constant-sized memory object (like in linear attention) but apply gradient

descent-like memory updates (Sun et al., 2024; Yang et al., 2025; Behrouz et al., 2025a; 2024; 2025b). Like our work, these architectures are motivated by the observation that gradient descent is very effective at compressing text into constant space and demonstrate the promise of using gradient descent at test time for long-context tasks. In contrast with our work, these architectures need to be trained from scratch, they have not been validated on large scale models, and do not match the quality of attention on recall-intensive tasks (Arora et al., 2024; Behrouz et al., 2025a).

### B.3.4 ORCHESTRATION FOR LONG-CONTEXT

In this section, we describe strategies for managing long-contexts by orchestrating calls to LLMs. For instance, the approach by (Russak et al., 2024) involves summarizing chunks of the context and then combining the summaries. Similarly, PRISM (Jayalath et al., 2024) treats the context as a sequence of chunks, capturing key information in a structured data format. MemGPT (Packer et al., 2023) introduces a virtual memory paging system, drawing inspiration from operating systems. As context length reaches the limit of available memory, the system strategically determines which information to retain.

### B.3.5 SYNTHETIC DATA GENERATION

A large body of work has focused on generating synthetic training data (Nayak et al., 2024; Abdin et al., 2024; Gandhi et al., 2024; Riaz et al., 2025). For example, Bonito is a model that is fine-tuned to generate synthetic data (Nayak et al., 2024), and MetaSynth is a method proposed by Riaz *et al.* that uses a language model to orchestrate several expert LLMs for domain-specific synthetic data generation (Riaz et al., 2025). The training process for Phi-4, a 14 billion parameter language model, also incorporates significant amounts of synthetically generated data (Abdin et al., 2024).

## C EXTENDED METHOD DESCRIPTION

In this section, we detail the seed prompts and chunking strategy we used to train CARTRIDGES with SELF-STUDY.

### C.1 SELF-STUDY SEED PROMPTS

As discussed in Algorithm 1, we seed the synthetic conversation generation with a prompt that elicits conversations about different aspects of the document. For each conversation, we randomly sample one of the following functions and create a seed prompt by calling it:

**Structuring Seed Prompt Generator**

```python
def structuring_seed_prompt(**kwargs):
    DATA_FORMATS = [
        "JSON",
        "YAML",
        "TOML",
        "INI",
        "XML",
        "plain text",
    ]

    data_format = random.choice(DATA_FORMATS)

    EXAMPLES = [
        (
            "Can you structure the information in {{subsection}} of {{document}} "
            "related to {{something specific}} "
            f"in the following format: {data_format}? "
            "Be sure to include precise information like any dates, times, names, and "
            "numerical values.'"
            ...
        )
    ]

    example = random.choice(EXAMPLES)

    return (
        f"Please generate a single chat message instructing an LLM to structure the "
        "information in {data_format}. "
        "Output only the chat message itself and absolutely nothing else. "
        "Make sure it is clear what section and document you are asking about. "
        f"The message can follow the following template, filling in details from the "
        "corpus: \n\n'{example}'"
    )
```

**Summarization Seed Prompt Generator**

```python
def summarization_seed_prompt(**kwargs):
    prompts = [
        (
            "Please generate a single chat message instructing an LLM to summarize "
            "part of the corpus. "
            "Make sure the instruction is very explicit about the section of the "
            "corpus that you want to summarize. "
            "Include details (ids, names, titles, dates, etc.) that make it clear what "
            "you are asking about. "
        ),
        (
            "Please generate a single chat message instructing an LLM to summarize a "
            "section. "
            "Make sure the instruction is explicit about the section that should be "
            "summarized and the document it is from."
        ),
    ]
    prompt = random.choice(prompts)
    return prompt
```

**Question Seed Prompt Generator**

```
1  def question_seed_prompt(**kwargs):
2      prompts = [
3          (
4              "Generate a question for an LLM that will test its knowledge of the
           information in the corpus above. "
5              "In your question be sure to include details (ids, names, titles, dates,
           etc.) that make it clear what you are asking about. "
6              "Output only a single question. Do NOT include any other text or
           explanation other than the question."
7          ),
8          (
9              "Generate a message for an LLM that will test its knowledge of the
           information in the corpus above."
10             "Be sure to include details (ids, names, titles, dates, etc.) in the
           question so that it can be answered without access to the corpus (i.e. closed-
           book setting). "
11             "Output only a single question. Do NOT include any other text or
           explanation other than the question."
12         ),
13         (
14             "You are helping to quiz a user about the information in the corpus. "
15             "Please generate a question about the subsection of the corpus above. "
16             "Be sure to include details (ids, names, titles, dates, etc.) in the
           question to make it clear what you are asking about. "
17             "Answer only with the question, do not include any other text."
18         ),
19     ]
20     prompt = random.choice(prompts)
21     return prompt
22
```

**Use Case Seed Prompt Generator**

```
1  def use_case_seed_prompt(**kwargs):
2      prompt = (
3          "You are working to train a language model on the information in the following
        corpus. "
4          "Your primary goal is to think about practical, real-world tasks or
        applications that someone could achieve using the knowledge contained within this
         corpus. "
5          "Consider how a user might want to apply this information, not just recall it.
        "
6          "After considering potential use cases, your task will be to generate a sample
         question that reflects one of these downstream applications. "
7          "This question/instruction/task should be something a user, who has access to
        this corpus, might ask when trying to accomplish their specific goal. "
8          "Output only a single question. Do NOT include any other text or explanation
        other than the question."
9      )
10     return prompt
11
12
```

**Creative Seed Prompt Generator**

```
1  def creative_seed_prompt(**kwargs):
2      prompt = [
3          (
4              "You are having a creative conversation inspired by the information in the
         corpus. "
5              "Please generate a question for your conversation partner to start off the
         discussion. "
6              "Answer only with the question, do not include any other text."
7          ),
8      ]
9      return random.choice(prompt)
```

## C.2   SELF-STUDY CHUNKING

For the SELF-STUDY data generation process, we extract uniformly random token-level chunks from the input corpus $\mathcal{C}$. A corresponding textual description is generally prepended to each chunk $\tilde{c}$ to contextualize it when generating the seed prompt. This approach helps the model focus on different parts of the corpus and generate diverse synthetic examples. The specific chunking parameters and descriptions are tailored to each dataset:

- **LONGHEALTH:** Chunks are sampled with a minimum size of 512 tokens and a maximum size of 4096 tokens. The accompanying description is: *'Below is a section of a patient's medical record. It is part of a larger corpus of medical records for $N_{patients}$ different patients.'*

- **AMD/FinanceBench:** Fixed-size chunks of 8192 tokens are utilized. No specific descriptive text is prepended to these chunks.

- **MTOB:** Chunks are sampled with a minimum size of 512 tokens and a maximum size of 4096 tokens. The description used is: *'The following is an excerpt from a grammar book about the Kalamang language.'*

- **QASPER:** Following our general methodology, chunks are sampled with a minimum size of 512 tokens and a maximum size of 4096 tokens. A generic description is used to contextualize the chunk as an excerpt from a research paper, in line with the nature of the Qasper dataset.

## D   DATASETS

### D.1   GENCONVO

To evaluate the ability of our approach to handle diverse queries over long documents, we generated the GENCONVO dataset. We created GENCONVO using the AMD 2022 10-K filing, a document from the FinanceBench corpus (Islam et al., 2023). The primary purpose of GENCONVO is to simulate a wide range of tasks a user might ask a model to perform given a long document, thereby testing the model's comprehension, reasoning, and ability to extract varied types of information. The generation process relies on Claude Sonnet 3.7 (Anthropic, 2024) and is structured as follows:

1. **Document Input:** The entire source document (e.g., the AMD 2022 10-K, which is less than 200,000 tokens and fits within the model's context window) is provided to Claude Sonnet 3.7.

2. **Question Generation:** A series of distinct prompt templates (detailed below), designed to elicit different reasoning traces (e.g., factual recall, synthesis, multi-hop reasoning), are used to generate questions. For the given document and each prompt template, we ask the model to generate 16 unique questions. This involves providing the model with the full document content alongside the specific question-generation prompt.

3. **Answer Generation:** Subsequently, for each generated question, Claude Sonnet 3.7 is prompted again with the original full document and the generated question to produce an answer. This process ensures that the answers are grounded in the provided document.

We hope GENCONVO provides a challenging benchmark that moves beyond simple fact retrieval, assessing a model's capacity for deeper understanding and more complex information processing over long contexts. The following prompt templates were utilized for the question generation phase:

**Factual Prompt Template**
```
Please generate a question to test someone's ability to remember
factual details from the document.  The answer should be a few tokens
long and be a factual detail from the statement, such as a number,
entity, date, title, or name.
This question should not be common knowledge:  instead, it should be
something that is only answerable via information in the document.
```

**Knowledge Prompt Template**

```
Please generate a question that requires combining information
mentioned both inside and outside the document.
This question should require using a fact from the document and
also a fact that you are confident about, but is not mentioned in
the document.  For instance:  - What are the founding dates of the
companies that got acquired this year?  This is a good question because
the names of the acquired companies are mentioned in the document
and the founding dates are not mentioned.  - What is the name of the
CEO's spouse?  This is a good question because the name of the CEO is
mentioned in the document and the spouse's name is not mentioned.
The answer should be a fact that is a few tokens long such as a number,
entity, date, title, or name.
```

**Disjoint Prompt Template**

```
Please generate a multi-hop question that tests someone's ability
to use factual information mentioned in at least two very different
sub-sections of the document.
This question shouldn't be a standard question about this kind of
document.  Instead, it should ask about two particularly disconnected
ideas, like comparing information about the amount of owned space
for the company headquarters with the amount of dollars of estimated
liability or comparing the revenue number with the number of employees.
This question should also test one's ability to do retrieval:  do not
give away part of the answer in the question.  Ensure that for one to
get the correct answer to the question, they need to understand the
document.
The answer should be a short:  for example, a number, entity, date,
title, or name.
```

**Synthesize Prompt Template**

```
Please generate a question that requires synthesizing and aggregating
information in the document.
For instance, you could ask someone to summarize a page of the
document, list all the key competitors mentioned in the document, or
summarize the company's business model.
```

**Structure Prompt Template**

```
Please generate a question that requires understanding the structure of
the document.
This question should be more about the structure of the document,
rather than the precise statement details.  For instance, you could
ask someone to list the titles of all the sections in the document,
describe the document structure, report the total number of pages, ask
which section amongst two sections comes first, or report the section
with the largest number of tables.
```

**Creative Prompt Template**

```
Please generate a question about the document to test someone's ability
to comprehend the content of the document.  This question specifically
should be focused on their ability to generalize the information about
the document to a strange question of sorts.
This question shouldn't be a standard question about this kind of
document, it should ask to do something abnormal and creative, like
writing a poem about a financial document.
```

---

**Counting Prompt Template**

```
Please generate a question that requires counting how frequently
different events occur in the document.
This question should be about statistical properties of the document,
rather than the statement details.  For instance, you could ask someone
to count the number of times the word "million" is mentioned or count
the length of the shortest section title.
The answer should be a number.
```

**Reasoning Prompt Template**

```
Please generate a question that requires mathematical reasoning over
the values in the document.
This question should require going beyond the facts directly mentioned
in the statement, such as asking to compute the percentage increase
in revenue between two years, find the largest expense category, or
calculate difference in profit between two years.
The answer should be a number.
```

## D.2 NEEDLE-IN-A-HAYSTACK (NIAH)

The Needle-in-a-Haystack task provides a controlled evaluation of a model's ability to precisely retrieve and recall specific information from long documents.

We adopt the challenging multi-key variant from the RULER benchmark (Hsieh et al., 2024), which requires models to locate and extract multiple pieces of information scattered throughout a long document. We choose this version of the task because it is more challenging than the standard single-key needle-in-the-haystack task and because it reflects the setting where CARTRIDGES are intended to be used: a single corpus of text against which many different queries are issued.

The task construction proceeds in three steps:

1. **Background Generation**: The document consists of random passages drawn from essays about startups by investor Paul Graham, creating realistic and semantically coherent text that serves as distracting context.
2. **Needle Insertion**: Multiple synthetic "needles" (key-value pairs) are inserted at random positions throughout the document. Each needle contains a unique identifier and an associated magic number. For example, the identifier "gorgeous-bath" is associated with the magic number "9290765".
3. **Query Formation**: LLLM prompts are produced that prompt the model to retrieve specific magic numbers given their corresponding identifiers, requiring precise information extraction from the long context. For example, the prompt "What is the magic number for gorgeous-bath?" requires the model to retrieve the magic number "9290765" from the long context.

This setup tests whether CARTRIDGES can maintain the same level of retrieval accuracy as ICL while using significantly compressed representations. The task is particularly challenging because the needles are syntactically similar but semantically distinct, requiring exact pattern matching rather than approximate retrieval.

Consider the excerpt below, which shows how needles are embedded within the natural text:

```
...  In the first couple weeks of working on their own startup they
seem to come to life, because finally they're working the way people
are meant to.Notes[1] When I talk about humans being meant or designed
to live a certain way, I mean by evolution.  [2] It's not only the
leaves who suffer.  The constraint propagates up as well as down.  So
managers are constrained too; instead of just doing things, they have
to act through subordinates.  One of the special magic numbers for
gorgeous-bath is:  9290765.  [3] Do not finance your startup with
credit cards.  Financing a startup with debt is usually a stupid move,
and credit card debt stupidest of all.  Credit card debt is a bad idea,
period.  It is a trap set by evil companies for the desperate and the
foolish.  ...
```

In this example, the model must identify that "gorgeous-bath" is associated with the magic number "9290765" when queried.

### D.3 LONGHEALTH

LONGHEALTH is a benchmark for evaluating large language models ability to analyze and interpret long clinical texts (Adams et al., 2024). The benchmark consists of 20 fictional clinical case reports (each containing between 5,090 and 6,754 word) and 400 multiple-choice questions based on them.

In our experiments, the context $\mathcal{C}$ consists of the reports for a *panel* of $n$ patients. We use $n = 10$ patients, with a full panel of approximately 100k tokens, which fits in the context length of the LLAMA 3 models.

The questions are categorized into information extraction, negation, and sorting.

A **sorting** question is included below:

```
Please answer the question below about the following patient:  ID
patient_03, Name:  Mr.  John Williams, Birthday:  1956-08-08 00:00:00,
Diagnosis:  Multiple Myeloma
<question>
Mr.  Williams received multiple radiologic examinations.  In which
order did she receive them?
</question>
<options>
CT Whole Body > MR Spine Scan > CT Spine Scan > PSMA-PET-CT Scan > CT
Chest > CT Whole Body > Whole Body CT scan
Whole Body CT scan > CT Spine Scan > CT Whole Body > MR Spine Scan > CT
Chest > PSMA-PET-CT Scan > CT Whole Body.
CT Whole Body > CT Whole Body > CT Chest > CT Chest > PSMA-PET-CT Scan
> MR Spine Scan > CT Spine Scan > Whole Body CT scan > Chest X-ray
CT Chest > CT Spine Scan > CT Whole Body > Whole Body CT scan >
PSMA-PET-CT Scan > MR Spine Scan > CT Whole Body
Whole Body CT scan > CT Spine Scan > CT Whole Body > MR Spine Scan > CT
Chest > CT Whole Body > PSMA-PET-CT Scan
</options>
You should first think step by step.  Then give your final answer
exactly as it appears in the options.  Your output should be in the
following format:
<thinking> {{YOUR_THOUGHT_PROCESS}} </thinking>

<answer>
{YOUR_ANSWER}
</answer>
```

An example of a **negation** question is included below:

```
Please answer the question below about the following patient:
ID patient_01, Name:  Anna Sample, Birthday:  1970-01-01
00:00:00, Diagnosis:  DLBCL
<question>
Which of these examinations were never performed in Mrs.
Sample?
</question>
<options>
Bone marrow aspiration
CSF aspiration
MRI of the head
Pulmonary function testing Cardiac stress testing
</options>
You should first think step by step.  Then give your final
answer exactly as it appears in the options.  Your output should
```

```
be in the following format:
<thinking> {{YOUR_THOUGHT_PROCESS}} </thinking>

<answer>
{YOUR_ANSWER}
</answer>
```

## D.4 MTOB

The Machine Translation from One Book (MTOB) benchmark tests a large language model's ability to learn to translate between English and Kalamang, a low-resource language with virtually no web presence (Tanzer et al., 2023). The core task is to perform translation (Kalamang to English, and English to Kalamang) by primarily relying on a single comprehensive grammar book and a small set of accompanying linguistic resources. In our work, we focus on translating from Kalamang to English.

The source documents provided by the MTOB benchmark are:

- **A grammar of Kalamang**: A comprehensive grammar textbook, with the original source provided in LaTeX format. This book details the phonology, morphology, and syntax of Kalamang.
- **Bilingual Word List (W)**: A list of Kalamang words with their part-of-speech tags and English descriptions.
- **Parallel Kalamang-English Corpus (S)**: A collection of 375 paired Kalamang-English sentences.

The MTOB authors preprocessed the grammar textbook from its original LaTeX source into several plaintext splits for their baseline experiments. These include:

- $G^m$ (**Medium-length chunk**): A plaintext segment of approximately 50k tokens consisting of an overview chapter, a morpheme table from the grammar book, and the complete bilingual word list (W).
- $G^l$ (**Long-length chunk**): A larger plaintext segment of approximately 100k tokens, containing chapters from the grammar book that the MTOB authors deemed most important for the translation task.
- **Full Plaintext Textbook (G)**: The entire grammar book converted to plaintext.

The combination of the long-length chunk ($G^l$), the parallel sentences (S), and the word list (W) exceeds the context window of Llama 3 models. We use the medium-length chunk $G^m$ and the parallel sentence list $S$ as input for our ICL baseline.

## D.5 QASPER

QASPER is a benchmark for evaluating the ability of large language models to answer questions about scientific papers (Dasigi et al., 2021). To create a challenging multi-query long-context setting resembling the setup described in Section 2.2, we concatenate 16 papers all related to *QA NLP models* to form out corpus $\mathcal{C}$. In total, there are 78 questions about these 16 papers in the dataset, which we use as the queries $Q$.

Because the dataset only includes short answers and ground-truth spans containing evidence for each answer, we rewrite the answers in a longer, more conversational format using GPT-4.1 and use these as the targets when evaluating.

## E THEORETICAL ANALYSIS: RELATIONSHIP BETWEEN ATTENTION, LINEAR ATTENTION, AND CARTRIDGES

When we generate text with an autoregressive Transformer, we have to maintain a KV-cache that grows linearly with the length of the input and text. In Section B.3.3, we discussed a number of architectural modifications that either reduce the size of the KV-cache or do away with it altogether.

In particular, when generating text with linear attention (*e.g.* (Arora et al., 2024)), we only need to maintain a constant-sized object – the KV-state matrix – during generation.

Like the KV-state matrix in linear attention, CARTRIDGES consume a constant amount of memory (*i.e.* their size is a hyperparameter, which can be set independently of the input length). However, they differ from the KV-state in how they are updated. In this work, CARTRIDGES are updated using SELF-STUDY– gradient descent on synthetically generated data. On the other hand, KV-states are updated using a linear attention update rule.

In this section, we will study the update rules for attention, linear attention, and gradient descent when applied to the multi-query associative recall (MQAR) problem (Arora et al., 2023), a popular synthetic benchmark task used for studying the capabilities of long-context architectures. In particular, we consider a variant of the standard MQAR problem where key-value pairs are repeated. First, we highlight some equivalences between the update rules of these approaches in the case where input keys are orthonormal. Then, in the more challenging case where input keys are in a Johnson-Lindenstrauss embedding, we provide a separation result showing that the gradient descent update rule is able to exactly solve an MQAR problem that linear attention cannot.

These theoretical results provide intuition for why constant-sized CARTRIDGES are able to match the performance of full KV-caches in long-context settings when linear-attention architectures have struggled to do so.

### E.1 NOTATION

All vectors are assumed to be row vectors.

Parenthesized superscripts (e.g. $\boldsymbol{k}^{(1)}$) denote some temporal quality of an element. Subscripts denote different elements in a set, as is standard.

A concise explanation for each variable:

- $d$ : model (and token) dimension.

- $m$ : number of unique key-value pairs.

- $n$ : number of queries.

- $N$ : number of key-value pairs in stream.

### E.2 MQAR

We define the Multiple Query Associative Recall (MQAR) problem.

**Definition 1.** *There is a universe of keys:*

$$K \subset \mathbb{R}^{1 \times d},$$

*and values:*

$$V \subset \mathbb{R}^{1 \times d}.$$

**Definition 2.** *(Arora et al., 2023) In the MQAR problem, the input is:*

$$(\boldsymbol{k}^{(1)}, \boldsymbol{v}^{(1)}), \ldots, (\boldsymbol{k}^{(N)}, \boldsymbol{v}^{(N)}) \text{ where } (\boldsymbol{k}^{(t)}, \boldsymbol{v}^{(t)}) \in K \times V \text{ for } 1 \leq t \leq N,$$

*followed by a set of queries*

$$\boldsymbol{q}_1, \ldots \boldsymbol{q}_n \text{ where } \boldsymbol{q}_i \in K \text{ for } 1 \leq i \leq n.$$

*Then for each $i \in [n]$, output:*

$$\begin{cases} \boldsymbol{v}_{i^*} \text{ where } i^* = \max\{i \in [1, N] | \boldsymbol{k}_i = \boldsymbol{q}_j\} \\ \mathbf{0}^d \text{ if no such } i \text{ exists.} \end{cases}$$

### E.3 m − REPETITIVE MQAR

**Definition 3.** *m − repetitive MQAR is a special case where each $(K^{(t)}, V^{(t)}) \in S$, where:*

$$S = \{(\boldsymbol{k}_1, \boldsymbol{v}_1), \ldots, (\boldsymbol{k}_m, \boldsymbol{v}_m)\}.$$

*Additionally, $\boldsymbol{k}_i$ is unique.*

**Definition 4.** *To capture this, $r_i^{(t)}$ is defined as the number of occurrences of $(\boldsymbol{k}_i, \boldsymbol{v}_i)$ in the stream at timestep $t$.*

#### E.3.1 ORTHONORMAL EMBEDDING

First, we will look at the MQAR problem in a restricted case, when all keys are orthonormal.

**Definition 5.** *We call the set $K$ to be orthonormal if for all $\boldsymbol{k}, \boldsymbol{k}' \in K$:*

$$\langle \boldsymbol{k}, \boldsymbol{k}' \rangle = \begin{cases} 0 & \text{if } \boldsymbol{k} \neq \boldsymbol{k}' \\ 1 & \text{otherwise.} \end{cases}$$

#### E.3.2 JOHNSON-LINDENSTRAUSS EMBEDDING

Next, we will look at the MQAR problem in a restricted case, when all keys are in a JL embedding.

**Definition 6.** *Let $\epsilon > 0$, we call the set $K$ to be $\epsilon−JL$ if for all $\boldsymbol{k}, \boldsymbol{k}' \in K$:*

$$\langle \boldsymbol{k}, \boldsymbol{k}' \rangle = \begin{cases} [-\epsilon, \epsilon] & \text{if } \boldsymbol{k} \neq \boldsymbol{k}' \\ 1 & \text{otherwise.} \end{cases}.$$

### E.4 MODEL DEFINITIONS

Below, we will describe three different model architectures. While they each exhibit different performance and capabilities they can be describe with a common framework for the MQAR problem.

1. State: is how the model store Key-Value pairs.

2. Update rule: how the model incorporates new Key-Value pairs into its state.

3. Query rule: how the model uses its state to answer a look up a value or a query.

#### E.4.1 TRANSFORMER

1. The state is:
$$\boldsymbol{W}^{(t)} = (\boldsymbol{K}^{(t)}, \boldsymbol{V}^{(t)}),$$

   where,
$$\boldsymbol{K}^{(t)} \in \mathbb{R}^{t \times d}, \boldsymbol{V}^{(t)} \in \mathbb{R}^{t \times d}.$$

   Note that this consumes more memory as the context gets longer.

2. The update rule is:
$$\boldsymbol{K}^{(t+1)} = \boldsymbol{K}^{(t)} \oplus \boldsymbol{k}^{(t+1)}, \boldsymbol{V}^{(t+1)} = \boldsymbol{V}^{(t)} \oplus \boldsymbol{v}^{(t+1)}$$

3. On query $\boldsymbol{q} \in K$, return:
$$\boldsymbol{q} \left( \boldsymbol{K}^{(t)} \right)^{\top} \boldsymbol{V}^{(t)}.$$

These rules define the transformer setting for MQAR.

### E.4.2 LINEAR ATTENTION

1. The state:
$$\boldsymbol{W}^{(t)} \in \mathbb{R}^{d \times d}.$$

2. The update rule is defined as:
$$\boldsymbol{W}^{(t+1)} = \boldsymbol{W}^{(t)} + (\boldsymbol{k}^{(t+1)})^\top (\boldsymbol{v}^{(t+1)}).$$

   With the initial matrix being initialized to zeros. I.e. $\boldsymbol{W}^{(0)} = \boldsymbol{0}^{d \times d}$.

3. On query q, return:
$$\boldsymbol{q}\boldsymbol{W}^{(t)}.$$

**Lemma 1.** *(Yang et al., 2025) Linear attention rule emerges if we were to update using the loss function $-\boldsymbol{k}^{(t)} \boldsymbol{W}^{(t)} \boldsymbol{v}^t$.*

It is important to mention here that we are not using any kernels for linear attention. These rules define the linear attention setting for MQAR.

**Lemma 2.** *(Yang et al., 2025) $\boldsymbol{W}^{(t+1)} = \boldsymbol{W}^{(t)} - \left(\boldsymbol{k}^{(t)}\right)^\top \boldsymbol{k}^{(t)} \boldsymbol{W}^{(t)} + \left(\boldsymbol{k}^{(t)}\right)^\top \boldsymbol{v}^{(t)}$ is the update rule that emerges when we use the gradient descent loss function: $\frac{1}{2}\|\boldsymbol{k}^{(t)} \boldsymbol{W}^{(t)} - \boldsymbol{v}^{(t)}\|_2^2$.*

**Definition 7.**
$$\mathcal{L} = \frac{1}{2}\|\boldsymbol{k}^{(t)} \boldsymbol{W}^{(t)} - \boldsymbol{v}^{(t)}\|_2^2$$

*Proof.* In general, gradient descent has the update rule:
$$\boldsymbol{W}^{(t+1)} = \boldsymbol{W}^{(t)} - \eta \nabla_{\boldsymbol{W}^{(t)}}. \tag{3}$$

Taking the gradient of the loss function gives us:
$$\nabla_{\boldsymbol{W}} \frac{1}{2}\|\boldsymbol{k}^{(t)} \boldsymbol{W}^{(t)} - \boldsymbol{v}^{(t)}\|_2^2 = \left(\boldsymbol{k}^{(t)}\right)^\top (\boldsymbol{k}^{(t)} \boldsymbol{W}^{(t)} - \boldsymbol{v}^{(t)})$$
$$= \left(\boldsymbol{k}^{(t)}\right)^\top \boldsymbol{k}^{(t)} \boldsymbol{W}^{(t)} - \left(\boldsymbol{k}^{(t)}\right)^\top \boldsymbol{v}^{(t)}.$$

Using the above and choosing $\eta = 1$, we get for Equation (3)
$$\boldsymbol{W}^{(t+1)} = \boldsymbol{W}^{(t)} - 1\left(\left(\boldsymbol{k}^{(t)}\right)^\top \boldsymbol{k}^{(t)} \boldsymbol{W}^{(t)} - \left(\boldsymbol{k}^{(t)}\right)^\top \boldsymbol{v}^{(t)}\right)$$
$$= \boldsymbol{W}^{(t)} - \left(\boldsymbol{k}^{(t)}\right)^\top \boldsymbol{k}^{(t)} \boldsymbol{W}^{(t)} + \left(\boldsymbol{k}^{(t)}\right)^\top \boldsymbol{v}^{(t)}.$$

$\square$

### E.4.3 GRADIENT DESCENT

Gradient descent training on the cache. We look at the capability of this trained state on a certain input.

1. The state at time $t$ is defined as:
$$\boldsymbol{W}^{(t)} \in \mathbb{R}^{d \times d}.$$

2. The update rule which follows from Lemma 2:
$$\boldsymbol{W}^{(t+1)} = \boldsymbol{W}^{(t)} - \left(\boldsymbol{k}^{(t)}\right)^\top \boldsymbol{k}^{(t)} \boldsymbol{W}^{(t)} + \left(\boldsymbol{k}^{(t)}\right)^\top \boldsymbol{v}^{(t)}.$$

   With the initial matrix being initialized to zeros. I.e. $\boldsymbol{W}^{(0)} = \boldsymbol{0}^{d \times d}$.

3. On query q, return:
$$\boldsymbol{q}\boldsymbol{W}^{(t)}.$$

### E.4.4 ORTHONORMAL CASE

We now see how the three models perform on the $m - $ repetitive MQAR when $K$ is orthonormal.

**Transformer**

**Lemma 3.** *On every input to MQAR (even those for 1-rep-MQAR) the state of Transformer needs* $\Omega(Nd)$ *parameters.*

Intuitively, at each timestep, you will append $d$ parameters to the state. At timestep $t$ the model will have $td$ parameters.

**Linear attention**

**Theorem 1.** *Linear attention can solve repetitive MQAR for any $m \geq 1$ and orthonormal $K$, up to scaling (producing $r_i^{(t)} \boldsymbol{v}_i$ when $\boldsymbol{W}^{(t)}$ is queried with $\boldsymbol{k}_i$) and all keys being distinct with $O(d^2)$ parameters.*

*Proof.* We first prove that for any $t \geq 0$:

$$\boldsymbol{W}^{(t)} = \sum_{i'=1}^{m} r_{i'}^{(t)} \boldsymbol{k}_{i'}^{\top} \boldsymbol{v}_{i'}. \tag{4}$$

**Base Case:** Initially, $\boldsymbol{W}^{(0)} = \boldsymbol{0}^{d \times d}$. From this, we indeed have:

$$\boldsymbol{W}^{(0)} = \sum_{i'=1}^{m} r_{i'}^{(0)} \boldsymbol{k}_{i'}^{\top} \boldsymbol{v}_{i'},$$

since for all $i' \in [m]$:

$$r_{i'}^{(0)} = 0.$$

**Inductive hypothesis:** Assume that the state matrix at some arbitrary integer timestep $t$ is as claimed. I.e.:

$$\boldsymbol{W}^{(t)} = \sum_{i'=1}^{m} r_{i'}^{(t)} \boldsymbol{k}_{i'}^{\top} \boldsymbol{v}_{i'}.$$

**Inductive step:** If $(\boldsymbol{k}^{(j)}, \boldsymbol{v}^{(j)})$ appears at timestep $t+1$ the update rule will be:

$$\boldsymbol{W}^{(t+1)} = \boldsymbol{W}^{(t)} + (\boldsymbol{k}^{(t+1)})^{\top} \boldsymbol{v}^{(t)}$$
$$= \boldsymbol{W}^{(t)} + (\boldsymbol{k}_j)^{\top} \boldsymbol{v}_j$$

By the inductive hypothesis, we have that:

$$\boldsymbol{W}^{(t+1)} = \boldsymbol{W}^{(t)} + \boldsymbol{k}_j (\boldsymbol{v}_j)^{\top}$$
$$= \sum_{i'=1}^{m} r_{i'}^{(t)} \boldsymbol{k}_{i'}^{\top} \boldsymbol{v}_{i'} + \boldsymbol{k}_j (\boldsymbol{v}_j)^{\top}$$
$$= \sum_{i'=1}^{m} r_{i'}^{(t+1)} \boldsymbol{k}_{i'}^{\top} \boldsymbol{v}_{i'}.$$

The final step follows from the fact that $r_j^{(t+1)} = r_j^{(t)} + 1$ when $(\boldsymbol{k}^{(t+1)}, \boldsymbol{v}^{(t+1)}) = (\boldsymbol{k}_j, \boldsymbol{v}_j)$ and $r_i^{(t+1)} = r_i^{(t)}$ for all $i \neq j$.
The proof of Equation (4) is complete by induction.

Finally, it is the case that on query $\boldsymbol{k}_i$:

$$\boldsymbol{k}_i \boldsymbol{W}^{(t)} = \boldsymbol{k}_i \sum_{i'=1}^{m} r_{i'}^{(t)} \boldsymbol{k}_{i'}^{\top} \boldsymbol{v}_{i'}$$

$$= \sum_{i'=1}^{m} r_{i'}^{(t)} \boldsymbol{k}_i \boldsymbol{k}_{i'}^{\top} \boldsymbol{v}_{i'}$$

$$= \sum_{i' \neq i} r_{i'}^{(t)} \boldsymbol{k}_i \boldsymbol{k}_{i'}^{\top} \boldsymbol{v}_{i'} + r_i^{(t)} \boldsymbol{k}_i \boldsymbol{k}_i^{\top} \boldsymbol{v}_i$$

$$= \sum_{i' \neq i} r_{i'}^{(t)} \cdot 0 \cdot \boldsymbol{v}_{i'} + r_i^{(t)} \cdot 1 \cdot \boldsymbol{v}_i$$

$$= r_i^{(t)} \cdot \boldsymbol{v}_i,$$

as desired. In the above, the second last inequality follows from from Definition 5 and the fact that all $\boldsymbol{k}_i$ are distinct.

$O(d^2)$ parameters are needed as the matrix must have dimension $d \times d$ $\qquad\square$

### Gradient Descent

**Theorem 2.** *Gradient descent is able to exactly solve the $m - repetitive$ MQAR (produce $\boldsymbol{v}_i$ when $\boldsymbol{W}^{(t)}$ is queries with $\boldsymbol{k}_i$) with $O(d^2)$ parameters.*

*Proof.* Here we can handle repetitions because our update rule includes a "peel" term. This means it removes the current value stored under a key before updating it with a new value.

We will show by induction that for all $t \geq 0$:

$$\boldsymbol{W}^{(t)} = \sum_{i'=1}^{m} \mathbb{1}_{r_{i'}^{(t)} > 0} \cdot \boldsymbol{k}_{i'}^{\top} \boldsymbol{v}_{i'}.$$

**Base Case:** Initially, the cache matrix is set to all zeros. From this, naturally follows that:

$$\boldsymbol{W}^{(0)} = \sum_{i'=1}^{m} 0 \cdot \boldsymbol{k}_{i'}^{\top} \boldsymbol{v}_{i'},$$

since for all $i'$

$$r_{i'}^{(0)} = 0.$$

**Inductive hypothesis:** Assume that at some arbitrary timestep $t$, we have:

$$\boldsymbol{W}^{(t)} = \sum_{i'} \mathbb{1}_{r_{i'}^{(t)} > 0} \cdot \boldsymbol{k}_{i'}^{\top} \boldsymbol{v}_{i'}$$

**Inductive step:** If $(\boldsymbol{k}_\ell, \boldsymbol{v}_\ell)$ appears at timestep $t + 1$ the update will be:

$$\sum_{i=1}^{m} \mathbb{1}_{r_i^{(t+1)} > 0} \boldsymbol{k}_i^{\top} \boldsymbol{v}_i = \left( \sum_{i'=1}^{m} \mathbb{1}_{r_{i'}^{(t)} > 0} \boldsymbol{k}_{i'}^{\top} \boldsymbol{v}_{i'} \right) - \left( \sum_{i'=1}^{m} \mathbb{1}_{r_{i'}^{(t)} > 0} \boldsymbol{k}_\ell^{\top} \boldsymbol{k}_\ell \boldsymbol{k}_{i'}^{\top} \boldsymbol{v}_{i'} \right) + \boldsymbol{k}_\ell^{\top} \boldsymbol{v}_\ell$$

the second term reduces to just peeling the term relating to $\boldsymbol{k}_\ell$, if it exists, as all other inner products are 0,

$$= \left( \sum_{i'=1}^{m} \mathbb{1}_{r_{i'}^{(t)} > 0} \boldsymbol{k}_{i'}^{\top} \boldsymbol{v}_{i'} \right) - \left( \mathbb{1}_{r_\ell^{(t)} > 0} \cdot \boldsymbol{k}_\ell^{\top} \boldsymbol{v}_\ell \right) + \boldsymbol{k}_\ell^{\top} \boldsymbol{v}_\ell$$

$$= \left( \sum_{i' \neq \ell}^{m} \mathbb{1}_{r_{i'}^{(t)} > 0} \boldsymbol{k}_{i'}^{\top} \boldsymbol{v}_{i'} \right) + \boldsymbol{k}_\ell^{\top} \boldsymbol{v}_\ell$$

This replaces the value associated with $\boldsymbol{k}_\ell$ with the new value, while keeping everything else the same. This is the form that we want, as the only time we want to add a key if it is an new key.

Finally, it is the case that on query $\boldsymbol{k}_i$:

$$
\begin{aligned}
\boldsymbol{k}_i \cdot \boldsymbol{W}^{(t)} &= \boldsymbol{k}_i \cdot \left( \sum_{i'=1}^m \mathbb{1}_{r_{i'}^{(t)}>0} \boldsymbol{k}_{i'}^\top \boldsymbol{v}_{i'} \right) \\
&= \left( \sum_{i'=1}^m \mathbb{1}_{r_{i'}^{(t)}>0} \boldsymbol{k}_i \cdot \boldsymbol{k}_{i'}^\top \boldsymbol{v}_{i'} \right) \\
&= \mathbb{1}_{r_i^{(t)}>0} \cdot 1 \cdot \boldsymbol{v}_i \\
&= \mathbb{1}_{r_i^{(t)}>0} \cdot \boldsymbol{v}_i
\end{aligned}
$$

Again here a matrix of dimension $d \times d$ can store $d$ orthogonal vectors. Thus this requires, $O(d^2)$ parameters.

$\square$

### E.4.5 JL EMBEDDING

We now see how the 3 models perform on the $m-$ repetitive MQAR when $K$ is $\epsilon-$JL.

**Transformer**

**Lemma 4.** *On every input to MQAR (even those for 1-rep-MQAR) the state of Transformer needs* $\Omega(Nd)$ *parameters.*

We note that when $K$ is $\epsilon-$JL it is no longer possible to get the exact answer from query rule $\boldsymbol{k}_i \boldsymbol{W}^{(t)}$. Thus, we need to add a decoding step.

**Definition 8.** *The output decoding step is* $\boldsymbol{v}_{i^*}$ *where:*

$$
i^* = \arg \max_{i' \in [m]} \langle \boldsymbol{v}_{i'}, \boldsymbol{k}_i \boldsymbol{W}^{(t)} \rangle.
$$

**Definition 9.** *For all* $i, j \in [m]$, *define:*

$$
\epsilon_{i,j} = \langle \boldsymbol{k}_i, \boldsymbol{k}_j \rangle.
$$

**Linear Attention**

**Theorem 3.** *Linear attention (+ decoding as in Definition 8) is unable to solve even the* $2-$ *repetitive MQAR and each* $\boldsymbol{v}_i$ *being 1-hot encoding unless* $K$ *is* $\omega\left(\frac{1}{N}\right) -$JL.

*Proof.* Due to the agreeance between different keys, when querying for key $i$, there is noise from other keys returned along with the correct answer. While we can tolerate some error, this error scales with the number of times the model has seen a single key. Making it unfit for longer contexts, or contexts with many repeats.

First, note that the base case Equation (4) from Theorem 1 still holds. In general, this holds for all $K$.

Specifically, on query $\boldsymbol{k}_1$ we have:

$$
\boldsymbol{k}_1 \boldsymbol{W}^{(t)} = r_1^{(t)} \langle \boldsymbol{k}_1, \boldsymbol{k}_1 \rangle \boldsymbol{v}_1 + r_2^{(t)} \langle \boldsymbol{k}_1, \boldsymbol{k}_2 \rangle \boldsymbol{v}_2 = r_1^{(t)} \boldsymbol{v}_1 + r_2^{(t)} \epsilon_{1,2} \boldsymbol{v}_2.
$$

Now, consider an input to $2-$ repetitive MQAR such that

$$
r_1^{(t)} < r_2^{(t)} \epsilon_{1,2}.
$$

Note that in this case:

$$
r_1^{(t)} = \langle \boldsymbol{v}_1, \boldsymbol{k}_1 \boldsymbol{W}^{(t)} \rangle < \langle \boldsymbol{v}_2, \boldsymbol{k}_1 \boldsymbol{W}^{(t)} \rangle = r_2^{(t)} \epsilon_{1,2}
$$

and hence we output $v_2$ instead of $v_1$.

If the embedding was $\omega(\frac{1}{N}$ the number of repeats could not overcome the $\epsilon$ value.

$\square$

### Gradient Descent

**Theorem 4.** *Gradient descent (+ decoding as in Definition 8) is able to exactly solve $m -$ repetitive MQAR with $O(d^2)$ parameters for $\epsilon-JL$ $K$, as long as $\epsilon \leq \frac{1}{m^2(m-1)}$ and $\alpha < \frac{m-1}{m+1}$.*

*Proof.* We define:

$$C_{i,j}^{(t)}$$

to be the coefficient associated with $k_i^\top v_j$ in $W^{(t)}$. Specifically, let

$$W^{(t)} = \sum_{i=1}^{m} \sum_{j=1}^{m} C_{i,j}^{(t)} k_i^\top v_j \tag{5}$$

We will prove by induction that:

$$C_{i,j}^{(t)} = \mathbb{1}_{(k_i, v_j) \text{ has occurred}} + \Delta_{i,j}^{(t)} \tag{6}$$

where,

$$\left| \Delta_{i,j}^{(t)} \right| \leq \sum_{a=1}^{t} ((m-1)\epsilon)^a. \tag{7}$$

**Base Case:** Initially, the state is set to all zeros. From this, naturally follows that all of the $C_{i,j}^{(t)}$ are zero. I.e. Equation (6):

$$\Delta_{i,j} = 0.$$

**Inductive hypothesis:** Assume that all for some timestep $t$ and $1 \leq i, j \leq m$:

$$C_{i,j}^{(t)} = \mathbb{1}_{(k_i, v_j) \text{ has occurred}} + \Delta_{i,j}^{(t)},$$

where $\Delta_{i,j}^{(t)}$ satisfies Equation (7).

**Inductive Step:** If at timestep $t+1$ we are given $(\boldsymbol{k}_\ell, \boldsymbol{v}_\ell)$, from Equation (5) the update looks like:

$$
\boldsymbol{W}^{(t+1)} = \sum_{i=1}^{m}\sum_{j=1}^{m} C_{i,j}^{(t+1)} \boldsymbol{k}_i^\top \boldsymbol{v}_j
$$

$$
= \sum_{i'=1}^{m}\sum_{j'=1}^{m} C_{i',j'}^{(t)} \boldsymbol{k}_{i'}^\top \boldsymbol{v}_{j'} - \left( \sum_{i'=1}^{m}\sum_{j'=1}^{m} C_{i',j'}^{(t)} \boldsymbol{k}_\ell^\top \boldsymbol{k}_\ell \boldsymbol{k}_{i'}^\top \boldsymbol{v}_{j'} \right) + \boldsymbol{k}_\ell^\top \boldsymbol{v}_\ell
$$

$$
= \sum_{i'=1}^{m}\sum_{j'=1}^{m} C_{i',j'}^{(t)} \boldsymbol{k}_{i'}^\top \boldsymbol{v}_{j'} - \left( \sum_{i'=1}^{m}\sum_{j'=1}^{m} \epsilon_{\ell,i'} C_{i',j'}^{(t)} \boldsymbol{k}_\ell^\top \boldsymbol{v}_{j'} \right) + \boldsymbol{k}_\ell^\top \boldsymbol{v}_\ell
$$

change the associativity of the summations,

$$
= \sum_{i'=1}^{m}\sum_{j'=1}^{m} C_{i',j'}^{(t)} \boldsymbol{k}_{i'}^\top \boldsymbol{v}_{j'} - \left( \sum_{j'=1}^{m}\left( \sum_{i'=1}^{m} \epsilon_{\ell,i'} C_{i',j'}^{(t)} \right) \boldsymbol{k}_\ell^\top \boldsymbol{v}_{j'} \right) + \boldsymbol{k}_\ell^\top \boldsymbol{v}_\ell
$$

here we separate the first term where $i' = \ell$ and $i' \neq \ell$,

$$
= \sum_{i'\neq\ell}\sum_{j'=1}^{m} C_{i',j'}^{(t)} \boldsymbol{k}_{i'}^\top \boldsymbol{v}_{j'} + \sum_{j'=1}^{m} C_{\ell,j'}^{(t)} \boldsymbol{k}_\ell^\top \boldsymbol{v}_{j'} - \left( \sum_{j'=1}^{m}\left( \sum_{i'=1}^{m} \epsilon_{\ell,i'} C_{i',j'}^{(t)} \right) \boldsymbol{k}_\ell^\top \boldsymbol{v}_{j'} \right) + \boldsymbol{k}_\ell^\top \boldsymbol{v}_\ell
$$

here we separate the first term where $i' = \ell$ and $i' \neq \ell$,

$$
= \sum_{i'\neq\ell}\sum_{j'=1}^{m} C_{i',j'}^{(t)} \boldsymbol{k}_{i'}^\top \boldsymbol{v}_{j'} + \sum_{j'=1}^{m} C_{\ell,j'}^{(t)} \boldsymbol{k}_\ell^\top \boldsymbol{v}_{j'} - \left( \sum_{j'=1}^{m} \epsilon_{\ell,\ell} C_{\ell,j'}^{(t)} \boldsymbol{k}_\ell^\top \boldsymbol{v}_{j'} \right) - \left( \sum_{j'=1}^{m}\left( \sum_{i'\neq\ell} \epsilon_{\ell,i'} C_{i',j'}^{(t)} \right) \boldsymbol{k}_\ell^\top \boldsymbol{v}_{j'} \right) + \boldsymbol{k}_\ell^\top \boldsymbol{v}_\ell
$$

remove $\epsilon_{j,j}$,

$$
= \sum_{i'\neq\ell}\sum_{j'=1}^{m} C_{i',j'}^{(t)} \boldsymbol{k}_{i'}^\top \boldsymbol{v}_{j'} + \sum_{j'=1}^{m} C_{\ell,j'}^{(t)} \boldsymbol{k}_\ell^\top \boldsymbol{v}_{j'} - \sum_{j'=1}^{m} C_{\ell,j'}^{(t)} \boldsymbol{k}_\ell^\top \boldsymbol{v}_{j'} - \left( \sum_{j'=1}^{m}\left( \sum_{i'\neq\ell} \epsilon_{\ell,i'} C_{i',j'}^{(t)} \right) \boldsymbol{k}_\ell^\top \boldsymbol{v}_{j'} \right) + \boldsymbol{k}_\ell^\top \boldsymbol{v}_\ell
$$

cancel terms,

$$
= \sum_{i'\neq\ell}\sum_{j'=1}^{m} C_{i',j'}^{(t)} \boldsymbol{k}_{i'}^\top \boldsymbol{v}_{j'} - \left( \sum_{j'=1}^{m}\left( \sum_{i'\neq\ell} \epsilon_{\ell,i'} C_{i',j'}^{(t)} \right) \boldsymbol{k}_\ell^\top \boldsymbol{v}_{j'} \right) + \boldsymbol{k}_\ell^\top \boldsymbol{v}_\ell.
$$

Note with this we can see that:

$$
C_{i,j}^{(t+1)} = \begin{cases} C_{i,j}^{(t)} & \text{if } \ell \neq i \\ -\sum_{i'\neq\ell} \epsilon_{\ell,i'} C_{i',j}^{(t)} + \mathbb{1}_{j=\ell} & \text{if } \ell = i \end{cases}.
$$

Thus, if $i \neq \ell$, we have:

$$
C_{i,j}^{(t+1)} = C_{i,j}^{(t)},
$$

for $i \neq \ell$. The inductive statement holds for these pairs. Now let's consider $C_{\ell,j}^{(t+1)}$. If $\ell = j$ then:

$$
C_{\ell,\ell}^{(t+1)} = 1 + \Delta_{\ell,\ell}^{(t+1)} = \sum_{i'\neq\ell} \epsilon_{\ell,i'} C_{i',j}^{(t)} + 1
$$

and note that by the triangle inequality and Definition 6:

$$\left|\Delta_{\ell,\ell}^{(t+1)}\right| \le \epsilon \sum_{i' \ne \ell} \left|C_{i',\ell}^{(t)}\right|$$

by the inductive hypothesis,

$$\le \epsilon \sum_{i' \ne \ell} (1 + \sum_{a=1}^{t} ((m-1)\epsilon)^a)$$

$$= ((m-1)\epsilon)(1 + \sum_{a=1}^{t} ((m-1)\epsilon)^a)$$

$$= (\sum_{a=1}^{t+1} ((m-1)\epsilon)^a),$$

as desired.

Then for $j \ne \ell$, we have:

$$\left|\Delta_{j,\ell}^{(t+1)}\right| = \left|C_{i,j}^{(t+1)}\right|$$

$$= \left|\sum_{i' \ne \ell} \epsilon_{\ell,i'} C_{i',j}^{(t)}\right|$$

The bounding of $\Delta_{\ell,j}^{(t)}$ is similar to the $\ell = j$ case.

With this we have completed the inductive proof on error terms.

If the we set:

$$\epsilon < \frac{1}{m^2(m-1)},$$

we get the following bound:

$$\Delta_{i,j}^{(t)} \le \sum_{a=1}^{t} ((m-1)\epsilon)^a \qquad (8)$$

$$\le \frac{(m-1)\epsilon}{1-(m-1)\epsilon} \qquad (9)$$

$$< \frac{1}{m^2-1} \qquad (10)$$

Before the next steps, we must bound:

$$|\langle \boldsymbol{v}_i, \boldsymbol{v}_j \rangle| \le \alpha \qquad (11)$$

For a query with $\boldsymbol{k}_i$, assuming we have seen $\boldsymbol{k}_i$ before, we get:

$$\boldsymbol{k}_i \cdot \boldsymbol{W}^{(t)} = \boldsymbol{v}_i + \sum_{j' \ne i} \Delta_{i,j'}^{(t)} \boldsymbol{v}_{j'}$$

Now for the decoding step where for an arbitrary $\boldsymbol{v}_j$ we get:

$$\langle \boldsymbol{v}_j, \boldsymbol{k}_i \cdot \boldsymbol{W}^{(t)} \rangle = \langle \boldsymbol{v}_j, \boldsymbol{v}_i \rangle + \langle \boldsymbol{v}_j, \sum_{j' \ne i} \Delta_{i,j'} \boldsymbol{v}_{j'} \rangle$$

For the case where $i = j$ it is the case that:

$$\langle \boldsymbol{v}_i, \boldsymbol{k}_i \cdot \boldsymbol{W}^{(t)} \rangle = 1 + \langle \boldsymbol{v}_i, \sum_{j' \neq i} \Delta_{i,j'} \boldsymbol{v}_{j'} \rangle$$

$$\geq 1 - \frac{1}{m+1} \alpha.$$

This follows from Equation (10) and Equation (11).

For the case where $i \neq j$ it is the case that:

$$\langle \boldsymbol{v}_j, \boldsymbol{k}_i \cdot \boldsymbol{W}^{(t)} \rangle = \langle \boldsymbol{v}_i, \boldsymbol{v}_j \rangle + \langle \boldsymbol{v}_j, \sum_{j' \neq i} \Delta_{i,j'} \boldsymbol{v}_{j'} \rangle$$

$$\leq \alpha + \frac{1}{m+1} \alpha$$

This follows from Equation (10) and Equation (11).

As a result, we will always pick the correct value when $\alpha < \frac{m-1}{m+1}$. $\qquad \square$

