# OpenReview forum: "Cartridges: Lightweight and general-purpose long context representations via self-study"
_ICLR.cc/2026/Conference — ICLR 2026 Poster_

### Official Review · Reviewer_AHXY · 2025-10-19

**Soundness:** 3
**Presentation:** 2
**Contribution:** 3
**Rating:** 6
**Confidence:** 2

**Summary:**

This paper introduces Cartridges, a memory-efficient, high-throughput, and general-purpose solution for processing long contexts in LLMs, promising to replicate the functionality of in-context learning (ICL). The authors argue that existing LLMs rely on costly KV caches for long-context processing or ICL. While KV cache compression can reduce memory, its performance collapses at high compression ratios.

To this end, they propose Cartridges, a lightweight, parametric cache that is trained offline to represent the text corpus. Direct optimization with the next-token prediction objective yields poor generalization, so they introduce self-study, training on synthetic conversational histories with a context-distillation objective.

On tasks such as NIAH, LongHealth, and MTOB, Cartridges achieves strong accuracy while reducing memory by 38.6× on average and boosting throughput by 26.4×, significantly outperforming state-of-the-art cache-compression baselines.

**Strengths:**

1. Long context is a critical research area for current LLMs, especially when it comes to both the ability to enhance the LLMs' own capabilities by processing ultra-long context as well as their efficiency.
2. Cartridge, as an architectural innovation, breaks away from the traditional KV-cache optimization paradigm, and its self-study training can also inspire future long-context training research.
3. The proposed method delivers extreme efficiency gains (38.6× memory reduction and 26.4× throughput boost) on very long contexts (up to 484k tokens) while still maintaining superior performance.

**Weaknesses:**

1. I would appreciate a better organization of the related work, instead of scattering it across Section 2.1, Appendix B, and Appendix E. From my perspective, it is a paper relatively focused on long-context efficiency, so it should be more concentrated on the comparison between cache optimization and linear attention. The key theoretical insights that appear only in Appendix B and E should also be highlighted in the main text. Although linear attention does require pre-training from scratch and non-trivial model conversions, a direct discussion of their trade-offs, whether in approach difference or in experimental performance, is essential for clarifying your methods.
2. The authors should more clearly distinguish Cartridges from prefix-tuning methods, for both learn a parametric cache, so a detailed explanation of their methodological distinctions and an experimental comparison are needed.
3. The paper reports results only in the Llama Series. Because Cartridges is presented as a general paradigm, additional experiments on Mistral and Qwen would strengthen the claim.
4. The authors claim that Cartridges can replicate the functionality of ICL; this should be backed better with evaluation on LongICLBench[1] or similar scalability tests in [2]. Likewise, their stated ability to support ultra-long contexts should be corroborated with results on RULER [3].


[1] LongICLBench: Long-context LLMs Struggle with Long In-context Learning https://arxiv.org/abs/2404.02060

[2] Many-Shot In-Context Learning https://arxiv.org/abs/2404.11018

[3] RULER: What's the Real Context Size of Your Long-Context Language Models? https://arxiv.org/abs/2404.06654

**Questions:**

See Weaknesses.

---

> ### Author Response · Authors · 2025-11-26
> **Author Response**
>
> Thank you for taking the time to review our work!
>
> > The paper reports results only in the Llama Series. Because Cartridges is presented as a general paradigm, additional experiments on Mistral and Qwen would strengthen the claim.
> >
>
> Please see Table 2, where we provide results on the Qwen3 family of models. The compression ratios achieved with Qwen are actually larger than those we achieved with Llama: (46.9x on MTOB and 106.4x on LongHealth).
>
> > *I would appreciate a better organization of the related work*.
> >
>
> Thank you for the suggestion! Our full discussion (Appendix B) is more than five pages long, so we could not include it in the main paper while respecting ICLR page limits. Section 2.1 is our effort to succinctly cover all of the work related to Cartridges. We felt it was important to provide readers with this extensive discussion detailing how our work relates to prior work, which is why we decided to include the full discussion in the appendix.
>
> > *The key theoretical insights that appear only in Appendix B and E should also be highlighted in the main text.*
> >
>
> In the revised manuscript, we now highlight the theoretical insights in Section 2.
>
> > *The authors should more clearly distinguish Cartridges from prefix-tuning methods, for both learn a parametric cache, so a detailed explanation of their methodological distinctions and an experimental comparison are needed.*
> >
>
> Thank you for the feedback! We’ve improved the clarity of Section 3.2, which details the relationship between our work and prefix tuning ([Li & Liang, 2021](https://arxiv.org/abs/2101.00190)). In short, in our work we parameterize Cartridges with a KV-cache, which is equivalent to vanilla prefix-tuning. So, the novelty of our work lies not the parameterization, which is equivalent to prefix-tuning, but in the finding that prefix-tuning combined with self-study can enable huge compression ratios relative to the full KV cache without degrading quality or generality. No prior work has shown these results.
>
> We justify the choice of the vanilla prefix-tuning parameterization and initialization in Figures 5 and 8. In Figure 8, we show experimentally how initializing with the KV cache of the first $k$ tokens of the corpus improves performance over prefix tuning with random initialization. In Figure 5, we show that prefix-tuning parameterization is more effective than a memory-matched LoRA
> parameterization on both in-domain and out-of-domain queries.
>
> > *Their stated ability to support ultra-long contexts should be corroborated with results on RULER [3].*
> >
>
> Thank you for the suggestion! In the updated manuscript, we have included results on synthetic tasks from the RULER framework. We focus on synthesizing RULER tasks that reflect the intended use case for Cartridges: settings in which many queries reference one context. The three tasks we evaluate are: Multi-Key Multi-Query Single-value NIAH, Multi-Key Multi-Query Single-value NIAH, and Variable Tracking. Note: that for all of our RULER evaluations, we did not change the **generic** **seed prompts** from the ones we used in the evaluations from the original paper.  We ran all of the experiments below with Llama 3.2 3B at 94k context length.
>
> The results for these experiments are now provided in Appendix A.5.
>
> *Multi-Key Multi-Query Single-value NIAH.*
>
> On multi-key, single-value NIAH, Cartridges achieves near perfect accuracy (> 99%), while compressing the context up to 92x.  Further, we performed a qualitative evaluation of the Cartridges trained on the NIAH corpus (at 92x compression). The model with the Cartridge can summarize the background “haystack” context (a random assortment of essays from the investor Paul Graham):
>
> > *User: In a sentence, can you summarize what you learned about?*
>
> > *Assistant: The text discusses various topics including the challenges of designing a language that programmers will love, … and the value of learning from others, including hackers and experts in the field.*
>
> We also checked that the model is capable of extracting the needle (`abundant-young=*9375247`)* when prompted in non-standard ways:
>
> > *User: What would the output of the following code be `print(magic_numbers["abundant-young"])`?*
>
> *> Assistant: The output of the code `print(magic_numbers["abundant-young"])` would be `9375247`.*
>
>
> *Multi-Key Multi-Query Multi-value NIAH.*
>
> The configuration of NIAH, in which the model has to output multiple values per key, is more challenging than the single-value variant above. The full KV-cache achieves only 28.8% exact match accuracy, which Cartridges can match with 23x compression.
>
> Note: that the original Ruler paper evaluates only with recall (instead of exact match). We found that the model often had high recall but low precision, so we report exact match to reflect this.
>
> *Variable tracking*
>
> On variable tracking, Cartridges outperforms the full KV cache (26.2% vs. 22.8% accuracy) while being 45.9x smaller.

---

### Official Review · Reviewer_mY8p · 2025-10-28

**Soundness:** 3
**Presentation:** 3
**Contribution:** 3
**Rating:** 6
**Confidence:** 4

**Summary:**

The work attacks the problem of high computational cost and reduced throughput of LLMs queried repeatedly to analyse a large (common) corpus of in-context documents.

The authors defined a cartridge - a sequence of pairs of pxd matrices that represent KV caches at every layer of the language model at hand. Here, p is the hyperparameter of the size of a cartridge, and d is the dimensionality of the model.

They suggest treating a cartridge as training parameters to represent information about a large common prefix. Given a common prefix, they suggest:
1. Generate a training dataset. In a loop, do:

1.a Retrieve a random chunk from the prefix

1.b. Seed a generic seed prompt from template categories (structuring, summarization, question, use case, creative)

1.c Generate a synthetic user query using the context of prefix chunk  + seed prompt

1.d Generate a synthetic assistant answer using the context of prefix chunk + synthetic user query

2. Construct a dataset of prefix chunk + synthetic user query + synthetic assistant answer
3. By minimizing the KL divergence between next-token predicted distributions produced by the model conditioned with the prefix chunk and the model conditioned with the cartridge over the parameters in the cartridge, we recover the optimal cartridge

Evaluations of LLaMa3-8B and Qwen3-4B on NIAH, LONGHEALTH, MTOB, and QASPER show very little quality degradation when using the cartridge instead of the long common prefix, despite ∼38.6× less memory and ∼26.4× higher throughput.

**Strengths:**

- The paper makes a clear, practical contribution. The proposed method has a high potential. Using self-study with chunking, the approach handles corpora beyond the model’s window. Cartridges can be concatenated at inference time. They are easy to serve with the existing infrastructure.
- The paper offers useful design ablations, including parameterization and initialization strategies.

**Weaknesses:**

The reproducibility of this paper is a potential weak point. Authors don’t provide code to reproduce their results, which may significantly diminish confidence and the potential impact.

Empirical results lack evaluations on real-world heterogeneous benchmarks, such as those with code or multimodal documents.

Operational procedures of versioning and updating cartridges are not discussed.

**Questions:**

The authors did not compare the QA performance of their cartridge system with the most obvious candidate, which is the RAG. Comparison with other prompt-compression methods (Lexico, Minicache, PALU, and KVPress) is also not provided. What would be the trade-off in accuracy and throughput?

Composition mechanism: The paper briefly notes that multiple CARTRIDGES can be composed at inference time, but the mechanism is unclear. Are there limitations in compositional depth or order sensitivity?


Evaluation breadth: The appendix lists several potential applications (e.g., summarization, retrieval, reasoning), yet the main results only cover QA and translation tasks. Were there any qualitative checks on these other use cases, even if not reported? If not, could you comment on any limitations that prevented such evaluations?

---

> ### Author Response · Authors · 2025-11-26
>
> Thank you for your review!
>
> > Empirical results lack evaluations on real-world heterogeneous benchmarks, such as those with code or multimodal documents.
> >
>
> Our empirical results do include results on real-world heterogenous benchmarks spanning medical reasoning, low-resource language translation, and scientific understanding. Multimodal cartridges that extend beyond prose to include images and code are an exciting direction for future inquiry, but beyond the scope of the present work. We discuss this limitation in the conclusion.
>
> > Operational procedures of versioning and updating cartridges are not discussed.
> >
>
> The Cartridges paradigm introduces operational challenges that practitioners would need to address such as versioning. However, studying particular approaches for versioning or updating is beyond the scope of this work, and would be an exciting direction for future inquiry.
>
> > The authors did not compare the QA performance of their cartridge system with the most obvious candidate, which is the RAG. Comparison with other prompt-compression methods (Lexico, Minicache, PALU, and KVPress) is also not provided.
> >
>
> We compare against a broad set of existing prompt and cache compression methods in our work. In Figure 3, we include comparisons with several compression baselines including a strong summary prompt compression method backed by GPT-5 and Duo-attention (the strongest cache compression method in [NVidia’s KVPress library](https://github.com/NVIDIA/kvpress). Further, in Table 1, we also include results for the next four best performing cache compression methods from KVPress.
>
> These comparisons demonstrate the substantive difference in compression enabled by Cartridges versus prior methods. Cartridges produces KV caches that are  up to 13.8× smaller for LongHealth, up to 97.0× for QASPER, and up to 648.3× for NIAH. In contrast, all of the cache compression baseline methods fail to match ICL quality even at modest compression ratios of 2−4×.
>
> Note that we do not compare against RAG, as is standard across the vast KV-cache compression literature (see, for example, [DuoAttention](https://arxiv.org/pdf/2410.10819), [Lexico](https://arxiv.org/html/2412.08890v1), [AdaKV](https://arxiv.org/pdf/2407.11550)). RAG is complementary to cache compression methods, since they increase the number of documents that can be retrieved and also can be used in conjunction.
>
> > Composition mechanism: The paper briefly notes that multiple CARTRIDGES can be composed at inference time, but the mechanism is unclear. Are there limitations in compositional depth or order sensitivity?
> >
>
> The composition mechanism is simply concatenation along the sequence dimension. We have added text to Section 5.4 explaining this. Note that composition is insensitive to order because we do not apply the positional encodings to the \artifact.
>
> > Evaluation breadth: The appendix lists several potential applications (e.g., summarization, retrieval, reasoning), yet the main results only cover Q/A and translation tasks.
> >
>
> Our main results do include evaluations of Cartridges on a diverse set of query types including summarization (Figure 2), mathematical reasoning (Figure 2), factual retrieval (Figure 3 left), data structuring tasks (Figure 2), creative tasks (Figure 2), translation (Figure 3 center right), medical reasoning (Figure 3 center left), and reading comprehension (Figure 3 right).
>
> > The reproducibility of this paper is a potential weak point. Authors don’t provide code to reproduce their results, which may significantly diminish confidence and the potential impact.
> >
>
> The code to reproduce the results will be made publicly available in an open source GitHub repository.

---

### Official Review · Reviewer_aS8F · 2025-11-01

**Soundness:** 3
**Presentation:** 4
**Contribution:** 3
**Rating:** 4
**Confidence:** 4

**Summary:**

This paper proposed a new method called self-study, which learns to store the document knowledge in the learnable KV cache parameters using a distillation loss; the authors propose to generate data augmentations based on the document to avoid training on the same document multiple times. Experiments show that self-study matches the performance of ICL with additional spend on training compute. In addition, this method enables the model to handle documents beyond its context length.

**Strengths:**

1. The paper proposed a new method called self-study, which learns to store the document knowledge in the learnable KV cache parameters. The method amortize the inference compute to training compute, which is useful in many real-world applications.
2. The authors carefully studied the effect of different initialization strategies, self-study compute, diversity of seed data augmentation prompts, etc., showing solid investigation on the factors that can impact the performance.
3. The work also studied different parameterization methods for the self-study, such as prefix-tuning and LoRA. The experimental ablations are comprehensive.

**Weaknesses:**

1. The Figures have very small font and is very hard to read.
2. The paper didn't compare with existing baselines on memory layer and active reading, nor discussing these works in the related work section, e.g., [1][2].
3. There is missing one strong baseline that a summarizer is used to condense the document in the text space before feeding long document to context.

[1] https://arxiv.org/abs/2412.09764
[2] https://arxiv.org/abs/2508.09494

**Questions:**

1. See Weaknesses. Could you provide comparison with the missing baselines?
2. What are the applications that you think will need such techniques the most--Do you envision the popular applications will have different parameters for different documents? In what case, the information loss due to compression will impact the model performance?

---

> ### Author Response · Authors · 2025-11-26
> **Author Response**
>
> Thank you for taking the time to provide feedback on our work!
>
>
> > There is missing one strong baseline that a summarizer is used to condense the document in the text space before feeding long document to context.
> >
>
> Our paper **does** compare against this baseline; see the `Summary` baseline in Figure 3. The numbers in the figure represent an upper bound on the performance of summary-based methods: we used the closed source GPT-5 models to generate the summaries (instead of the Llama/Qwen-3B/8B models). This advantages the summary baseline when comparing to Cartridges, which of course does not rely on any frontier models. Even though we significantly advantage the baseline, at equivalent compression ratios, Cartridges outperforms the summary upper bound by at least 10 accuracy points (LongHealth) and as much as 100 accuracy points (Multi-key NIAH).
>
> > What are the applications that you think will need such techniques the most--Do you envision the popular applications will have different parameters for different documents?
> >
>
> The memory reductions enabled by Cartridges could already have a practical impact on mainstream, real-world workloads. In many settings, a users’ long context is fixed or changes slowly, so the Cartridge can be trained offline and then quickly loaded when referenced by the user at query time. This would reduce the time-to-first token and increase tokens/sec, making working with long contexts cheaper, more instantaneous, and more interactive. For instance, a 76.6x memory reduction (which we achieve with Qwen) corresponds to >45x increase in tokens/second with Llama 8B at peak throughput .
>
> Below are some important examples of such settings:
>
> - *Code bases*. On a large software engineering team, users could issue thousands of queries in a code base per day (think of tab-complete in Cursor). Ideally, the LLM would have the entire code base in context when responding, but this is rarely done due to context limits and prohibitive cost. One could imagine making this cost-effective by training a Cartridge nightly and keeping recent diffs in the standard KV cache.
> - *Long-term memory for chatbots*. Chatbot providers (e.g. ChatGPT) would like to condition responses on all previous interactions they’ve had with a user so that the experience is more personalized. However, putting the entire chat history in context for every message is prohibitively expensive, so most providers rely on aggressive prompt compression. However, as we demonstrate in our work, these techniques often fail to replicate the behavior of the full KV cache. Instead, we could maintain one cartridge per user, since the chat history evolves slowly, and the cartridge could be updated nightly and used repeatedly across many user queries.
> - *Shared knowledge bases and wikis*. Organizations often keep institutional know-how in large knowledge bases like Google Drive or Notion. Ideally, users could include all of this information when chatting with an LLM about work. We could instead train a single cartridge on an organization’s knowledge base and share it among users in the organization.
>
> Though future work would need to iron-out domain-specific details for each of these settings, we believe our work presents a first step towards realizing these long-context applications in a cost-effective way.
>
> > In what case, the information loss due to compression will impact the model performance?
> >
>
> Cartridges suffer from information loss relative to the full KV cache in settings where we can only afford to spend a small amount of compute on self-study. In Figure 3 bottom, we demonstrate this. When we use small amounts of self-study compute, we underperform the full KV cache. On the other hand, if we use sufficient self-study compute, we can match or outperform the full KV cache while using dramatically less space (76.6x with Qwen and 38.6x with llama). Even on NIAH, a task explicitly designed to measure information loss, we can achieve perfect retention (*i.e.* no information loss).
>
> At first glance, this may sound like free lunch. To understand how this is possible, consider the analogy to a lossless compression algorithm like Huffman encoding, which uses a more costly encoding algorithm to represent data without any information loss. Similarly, by using gradient descent, a more costly encoding algorithm than prefill, self-study can produce a smaller KV cache with no information-loss.
>
> So, the takeaway from our paper is that as long as you are willing to spend enough training compute producing the Cartridge, we can retain the performance of the full KV cache, without information loss. If we cannot afford to spend compute training the Cartridge, then we suffer information loss (no free lunch).

---

> > ### Author Response · Authors · 2025-11-26
> > **Author Response Cont.**
> >
> > > The paper didn't compare with existing baselines on memory layer and active reading, nor discussing these works in the related work section, e.g., [1][2].
> > >
> >
> > Active reading proposes a synthetic data generation process, which is nearly equivalent to the synthetic data generation used in self-study. In active reading, the synthetic data generation is used to perform continued pretraining of the full model, whereas in our work, it is used to compress per-user KV caches.  Note that the active reading paper is also under review at ICLR 2026, and thus, is considered contemporaneous work under the [ICLR policy](https://iclr.cc/Conferences/2025/FAQ?utm_source=chatgpt.com). Nonetheless, we’ve highlighted it in the revised related work.
> >
> > The memory layers paper proposes a new LLM architecture that the authors pre-train from scratch. Unlike the prefix-tuning or LoRA papers, the memory layers paper does not propose a method for adapting a pre-existing Transformer language model, so it is not clear how it can be used to parameterize a Cartridge.  One could devise a way to integrate new memory layers into a pretrained Transformer, but developing such a method is beyond the scope of the present work.
> >
> > > The Figures have very small font and is very hard to read.
> > >
> >
> > Thank you for the feedback! We’ve increased the font in some of the figures.

---

### Official Review · Reviewer_RZFK · 2025-11-01

**Soundness:** 3
**Presentation:** 3
**Contribution:** 3
**Rating:** 6
**Confidence:** 4

**Summary:**

This work explores the challenge of efficiently serving large language models (LLMs) for tasks requiring access to extensive text corpora.
Traditional approaches rely on placing entire corpora in the model's context window, which is memory-intensive due to the scaling of key-value (KV) cache with input length.
The authors propose an alternative: pre-training a smaller, corpus-specific KV cache called a Cartridge offline, which can be reused across multiple queries referencing that corpus.
While naive next-token prediction training does not yield competitive results compared to standard in-context learning (ICL), the authors introduce a "self-study" method, generating synthetic conversations about the corpus and using context-distillation objectives, to train Cartridges effectively.
The experimental results show that self-study-trained Cartridges match ICL performance on long-context benchmarks while dramatically reducing memory usage and increasing throughput.
Additionally, this approach extends effective context length and allows for composability of Cartridges without retraining.

**Strengths:**

1. The self-study technique leverages synthetic data generation and distillation objectives to further improve context compression, enabling high-quality representation learning beyond naive next-token prediction.
2. The experimental results show that Cartridge significantly reduces memory consumption and increases throughput by over 26x compared to traditional ICL methods. Despite resource savings, Cartridge achieves comparable performance as full-context ICL on challenging benchmarks.
3. The writing clearly demonstrates the paper's motivation, implementation, and analysis.

**Weaknesses:**

1. Effectiveness relies heavily on generating high-quality synthetic conversations. There should have been some analysis on the data quality and cost of synthetic data.
2. It would be better to compare with some naive methods that have a similar idea with Cartridge, e.g., existing prompt compression methods + SFT with your synthetic data.

**Questions:**

See weakness.

---

> ### Author Response · Authors · 2025-11-26
> **Author Response**
>
> Thank you for taking the time to review our work!
>
> > *It would be better to compare with some naive methods that have a similar idea with Cartridge, e.g., existing prompt compression methods + SFT with your synthetic data.*
>
> **Baseline compression methods.**  We compare against a broad set of existing prompt and cache compression methods in our work. In Figure 3, we include comparisons with several compression baselines including a strong summary prompt compression method backed by GPT-5 and Duo-attention (the strongest cache compression method in [NVidia’s KVPress library](https://github.com/NVIDIA/kvpress). In Table 1, we also include results for the next four best performing cache compression methods from KVPress.
>
> These comparisons demonstrate the substantive difference in compression enabled by Cartridges versus prior methods. Cartridges produces KV caches that are  up to 13.8× smaller for LongHealth, up to 97.0× for QASPER, and up to 648.3× for NIAH. In contrast, all of the cache compression baseline methods fail to match ICL quality even at modest compression ratios of 2−4×.
>
> **SFT with synthetic data.** In Figure 4 (center-right panel), we compare self-study with context distillation to simple next-token prediction (*i.e.* SFT with the synthetic data). We find that the context-distillation objective in self-study outperforms the next-token prediction objective by 8.6 ChRF points on MTOB.
>
> > *Effectiveness relies heavily on generating high-quality synthetic conversations. There should have been some analysis on the data quality and cost of synthetic data.*
> >
>
> The quality of the Cartridge does indeed depend on (1) the distribution of synthetic conversations and (2) the number of synthetic conversations. We provide an analysis of how the data distribution affects Cartridge quality in Figure 4 (center-right panel). In Figure 2 bottom we provide analysis of how performance scales with number of synthetic conversations (*i.e.* cost).
>
> Below we provide an overview of some of the insights from these analyses.
>
> Recall, the model generates the synthetic conversation **autonomously**: we simply put the corpus (or chunks of it) into the system prompt, and let it chat back and forth with itself. We do not explicitly control the exact queries/requests the model makes.
>
> The one part we do control is the ***seed prompt***, a temporary chat message that we include only on the very first LLM request that instructs the model to generate a conversation starter. In Appendix Figure 9, we show that a single simple seed prompt already covers a large space of prompts (achieving strong performance on LongHealth and QASPER): “Please generate a single chat message to begin a conversation about the information in the corpus. Ask a question about the corpus or make a request.”
>
> However, as we show in Figure 9, we find that performance improves if we instead randomly sample a seed prompt from one of five types: structuring, use case, summarization, creative, question. Critically, the seed prompts we use are always generic: except for ablation experiments, we do not vary them from task-to-task. One can find all of the seed prompts in their entirety in Appendix B.1. For example, here is the creative seed prompt: "You are having a creative conversation inspired by the information in the corpus. Please generate a question for your conversation partner to start off the discussion."
>
> There is no way to guarantee that this set of seed prompts provides sufficient coverage over all prompts. However, empirically, we find that this concise set of five seed prompts generalizes to a broad set of prompts:
>
> - In Figure 2, we evaluate on diverse query types, including some that are not represented in the seed prompts (*e.g.* mathematical reasoning, disjoint reasoning).
> - We achieve parity on LongHealth, despite never mentioning **multiple choice** or **medicine** in the seed prompts.
> - We achieve parity on MTOB, despite never mentioning **translation** in the seed prompts.

---

### Author Response · Authors · 2025-11-26
**General Response**

Thank you for taking the time to review our manuscript and for providing detailed feedback and questions. We were pleased to see that the reviewers highlighted many of the strengths of the paper:

- “The paper makes a clear, practical contribution. The proposed method has a high potential.” [mY8p]
- The methodology enables “extreme” [AHXY]  efficiency gains on very long contexts by “significantly reducing memory consumption” [RZFK].
- The ablation study is “useful" [mY8p] and “comprehensive” [aS8F] providing a “solid investigation on the factors that can impact the performance” [aS8F].
- The writing “clearly demonstrates the paper's motivation, implementation, and analysis.” [RZFK]

The reviewers raised a number of questions, which we respond to in the reviewer-specific comments below and with revisions to the manuscript. Altogether, our revisions and responses provide further support for our paper’s central claim: *that offline training with self-study is an effective way to compress KV caches, when contexts are reused across many queries.*

---

### Meta-Review · Area_Chair_ZRQb · 2025-12-10

**Summary:**

This paper introduces approaches to effectively learn what authors refer to as cartridges: learned KV-cache-like representations of content one may want to generate conditionally on at test time. Crucially, cartridges can potentially be made much shorter than the actual cache, enabling faster inference due to reduced cache communication overhead. Authors show that naively training such a parametric cache will incur overfitting, which they address with an approach to generate synthetic data. The proposal is timely and practical, and results provide enough evidence supporting the authors's claims.

**Reviewer Concerns:**

Reviewers were mostly positive about the paper, and light concerns were raised regarding extra experiments related to assessing the quality of the synthetic data or somehow measuring information loss due to compressions. While those would be interesting additions, I would argue the paper does provide enough evidence to what's claimed.

**Reviewer Scores:**

I would not expect changes in scores given the discussion. The positive leaning reviewers only raised light concerns, and the negative leaning one asked for additional results that were not fully provided in the authors's responses, so I wouldn't expect a score change. Although I tend to side with the more positive reviewers as I do think the paper already provides enough results to sustain the relevance of the work.

---

### Decision · Program_Chairs · 2026-01-26

Accept (Poster)